



**Simulation characteristics of seismic translation and rotation**
**under the assumption of nonlinear small deformation**
Wei Li [1,2], Yun Wang [1,2,*], Chang Chen[1,2], Lixia Sun[1,3]
[1] *"MWMC" group, School of Geophysics and Information Technology, China*
*University of Geosciences, Beijing* 100083*, China*
[2] *State Key Laboratory of Geological Processes and Mineral Resources, China*
*University of Geosciences, Beijing* 100083*, China*
[3] *Sinopec Research Institute of Petroleum Engineering Co., Ltd., Beijing* 102206*,*
*China*
[*] Corresponding author: wangyun@mail.gyig.ac.cn.
**Abstract** The conventional theory of elastic-wave propagation is based on classical
elastodynamics, assuming linear small deformations of particles. However, recent
observations of seismic rotation have revealed significant disparities between actual
rotational motions induced by earthquakes in focal areas and near fields compared to
theoretical calculations and simulations. Considering the nonlinearity may be the
main cause of the discrepancies and based on classical elastodynamic principle, we
derive seismic elastic-wave equations with Green strain tensor without the linear
small deformation assumption, a different way from using complex nonlinear
constitutive relation and try to interpret the mechanism of seismic rotation. By
simulating and analyzing translational and rotational components subjected to the
three basic and typical vibrating sources, namely, isotropic (ISO), double couple (DC),





and compensated linear vector dipole (CLVD), represented by moment tensors, we
investigate the wavefield differences between elastic-wave equations based on linear
and nonlinear geometric relations and quantify the differences in homogeneous elastic
full-space model. Subsequently, we simulate two observed six-component Taiwan
earthquakes and compare their differences caused by nonlinear simulations. The
results indicate that linear approximation errors are more pronounced in seismic ISO
and CLVD sources. And the nonlinearity of small deformation has a more pronounced
effect on rotational motions deduced by strong earthquakes. Also, the nonlinear
mechanics of seismic rotation can attribute to the complex propagation paths and
source mechanisms simultaneously.



## 1 Introduction

Seismic rotational motions are recorded in plenty of earthquakes, especially in
strong shocks (Grayzer, 1991; Graizer, 2010; Zhou et al., 2019). Several studies have
concluded that rotational motions cannot be neglected in shallow foci and near-field
seismology (Kozak, 2009; Sun et al., 2017). In architecture engineering, rotational
torsions are encouraged to be considered in assessing the stability of ground motions
and building design (Li, 1991; Li and Sun, 2001; Yan, 2017; Huras et al., 2021).
Many studies suggest that including seismic rotation data, which records spatial
gradients, will enhance the precision of earthquake source prediction and moment
tensor inversion (Bernauer et al., 2014; Donner, 2016; Ichinose et al., 2021), as
validated in simulations by Hua and Zhang (2002).
Lee (2007) ever summarized the practical applications of observing seismic
rotations in engineering, attributing seismic rotation to nonlinear elasticity and site
effects. Notably, observed rotations during strong ground motions exceed calculated
translational components by one to two orders of magnitude. Recognizing the pivotal
role of nonlinear wave propagation in addressing geophysical complexities stemming
from Earth's heterogeneities, various analytical solutions of nonlinear wave equations
have been advanced through iterative techniques based on Green's function (McCall,
1994), including the flux-corrected transport method (Yang et al., 2002; Zheng et al.,
2006), and perturbation approaches (Bataille and Contreras, 2009; Jia et al., 2020) to
investigate the nonlinear effects on elastic waves. However, existing studies
predominantly concentrate on the nonlinear constitutive relations of stress and strain,



traditionally assuming linear small deformations (Renaud et al., 2012; Renaud et al., 2013b; TenCate et al., 2016; Feng et al., 2018), scarcely exploring nonlinearity in geometric relationship, which may be a crucial aspect that could better approximate strong rotational motions and near-field seismic conditions.

Taiwan, located in an active seismic region where earthquakes have garnered attention for their special rotational characteristics of distinctive strike-slip, particularly evident in the southern and northern areas, has been highlighted by extensive broadband seismic observations and earthquake-physical studies (Yu et al., 1999; Wang and Lv, 2006). Oliveira and Bolt's studies (1989) underscore the significant impact of rotation in near-field observations on the island, and Chen et al. (2014) discovered vertical rotations and frequency spectrum variations between horizontal and vertical rotations in the near zone of earthquakes from 2007 to 2008. These findings incline the importance of rotational studies in unraveling Taiwan's underground structures and geodynamics.

In this study, we first investigate the rotational characteristics under the assumption of nonlinear small deformation through numerical simulations of three basic seismic moment tensor sources. Additionally, we engage in theoretical simulations of six-component (6C) wavefields using observations from near and strong seismicity in Taiwan. We employ the Green strain tensor in the simulations of seismic wavefields to discuss the linear approximation and the earthquake mechanisms at play in this region.

**2 Theories**



2.1 Elastodynamic theory
In a three-dimensional orthogonal Cartesian coordinate system depicting an elastic
body within an elastic space, illustrated in Fig. 1, consider point **A** within the elastic
body, denoted as **x**, while point **B**, located in the immediate vicinity of **A**, is indicated
as **x**+d**x**. The infinitesimal distance between **A** and **B** is defined as *ds*. Under
instantaneous motivation of an external force, the elastic mass element **AB**
experiences displacement *u*(*x*, t), transitioning to a new position **A'B'**, followed by
small deformation of the elastic body, where the positions **A'** and **B'** are designated as
*x′* and *x′*+d*x′*, respectively, and their distance denoted as d*s′*. The work done by the
external force is primarily transformed into kinetic energy due to displacement and
potential energy stemming from elastic deformation. Hence, the change in square of
the length of a line element before and after its deformation is used to measure the
deformation, i.e., the squared difference in distance between **AB** and **A'B'**, expressed
by Eq. (1). The following equations and tensors are written using the Kronecker
symbol and dummy indicator rules.

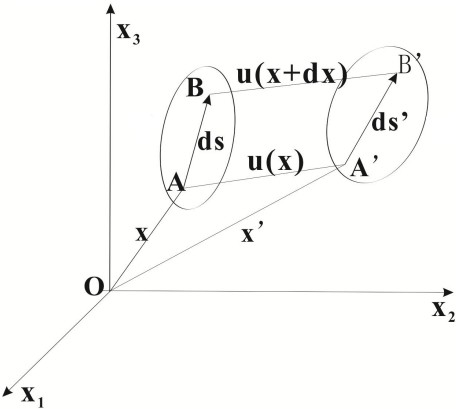


**Figure 1.** Schematic diagram of displacement and deformation of an elastomer



(Adapted from Aki and Richards (2002))
$$(ds')^2 - (ds)^2 = (\frac{\partial u_j}{\partial x_i} + \frac{\partial u_i}{\partial x_j} + \frac{\partial u_k}{\partial x_i} \cdot \frac{\partial u_k}{\partial x_j})dx_i dx_j \, , \text{ i, j=x, y, z} \tag{1}$$

Where $u_i$ and $u_j$ denote the displacements in different directions, and $x_i$ and $x_j$
denote the specific X, Y, and Z axes in Cartesian coordinates. Eq. (2), known as the
Green strain tensor, serves as an objective measure of the strain tensor before and
after the deformation of an elastomer.
$$E_{ij} = \frac{1}{2}(\frac{\partial u_j}{\partial x_i} + \frac{\partial u_i}{\partial x_j} + \frac{\partial u_k}{\partial x_i} \cdot \frac{\partial u_k}{\partial x_j}) \tag{2}$$

The strain ($e_{ij}$) and rotation ($r_{ij}$) tensors in elastodynamic theory are defined as:
$$e_{ji} = \frac{1}{2}(\frac{\partial u_j}{\partial x_i} + \frac{\partial u_i}{\partial x_j}) \tag{3}$$

$$r_{ji} = \frac{1}{2}(\frac{\partial u_i}{\partial x_j} - \frac{\partial u_j}{\partial x_i}) \tag{4}$$

Then, the Green strain tensor can be written as Eq. (5).
$$E_{ij} = e_{ij} + \frac{1}{2}e_{ij}^2 + \frac{1}{2}(e_{ij}r_{ij} - r_{ij}e_{ij}) - \frac{1}{2}r_{ij}^2 \tag{5}$$

The second-order displacement of nonlinearity in the Green tensor is neglected in
the classical theory of kinetic elasticity. Instead, it focuses solely on the first-order
linear terms, simplifying the nonlinear strain tensor to $e_{ij}$.
In small deformation assumption, the volumetric strain due to shear strain during
elastomer deformation is overlooked, shifting the focus solely to the volumetric strain
along the three principal stress axes (Eq. (6)).
$$\theta = \frac{(1+\theta_{xx})(1+\theta_{yy})(1+\theta_{zz})dxdydz - 1dxdydz}{dxdydz} \tag{6}$$
$$= e_{xx} + e_{yy} + e_{zz} + e_{xx}e_{yy} + e_{xx}e_{zz} + e_{yy}e_{zz} + e_{xx}e_{yy}e_{zz}$$



The simplified linear strain tensor $e_{ij}$, which ignores the actual nonlinear
displacement term in the Green strain tensor, retains only first-order linear terms in its
volumetric strain (Eq. (7)).

$$\theta_e \approx e_{xx} + e_{yy} + e_{zz} = \frac{\partial u_x}{\partial x} + \frac{\partial u_y}{\partial y} + \frac{\partial u_z}{\partial z} \tag{7}$$

The linear strain tensor, $e_{ij}$, disregards the nonlinearity present in the Green strain
tensor, and it becomes evident that only the first-order linear terms are retained in the
volumetric strain (Eq. (7)). The volumetric strain related to the Green strain tensor
features nonlinear second-order displacement terms while discounting higher-order
components (Eq. (8)).

$$\theta_E \approx E_{xx} + E_{yy} + E_{zz} + E_{xx}E_{yy} + E_{xx}E_{zz} + E_{zz}E_{yy} \approx \frac{\partial u_x}{\partial x} + \frac{\partial u_y}{\partial y} + \frac{\partial u_z}{\partial z} + \frac{\partial u_x}{\partial x}\frac{\partial u_y}{\partial y} + \frac{\partial u_x}{\partial x}\frac{\partial u_z}{\partial z} + \frac{\partial u_y}{\partial y}\frac{\partial u_z}{\partial z}$$
$$+ \frac{1}{2}\left( \frac{\partial u_x}{\partial x}\cdot\frac{\partial u_x}{\partial x} + \frac{\partial u_y}{\partial x}\cdot\frac{\partial u_y}{\partial x} + \frac{\partial u_z}{\partial x}\cdot\frac{\partial u_z}{\partial x} + \frac{\partial u_x}{\partial y}\cdot\frac{\partial u_x}{\partial y} + \frac{\partial u_y}{\partial y}\cdot\frac{\partial u_y}{\partial y} + \frac{\partial u_z}{\partial y}\cdot\frac{\partial u_z}{\partial y} + \frac{\partial u_x}{\partial z}\cdot\frac{\partial u_x}{\partial z} + \frac{\partial u_y}{\partial z}\cdot\frac{\partial u_y}{\partial z} + \frac{\partial u_z}{\partial z}\cdot\frac{\partial u_z}{\partial z} \right) \tag{8}$$

By combining $e_{ij}$, called the geometric equation, with the linear elastic constitutive
equation given by Hooke and Cauchy equations, the conventional elastic-wave Navier
equation (Eq. (9)) is obtained, which represents the linear elastic-wave equation
within the realm of isotropic media, premised on the assumption of linear small
deformations.

$$\rho \frac{\partial^2 u_i}{\partial t^2} = \rho f_i + (\lambda + \mu)\frac{\partial \theta}{\partial x_i} + \mu \frac{\partial^2 u_i}{\partial x_j \partial x_j} \tag{9}$$

Where $\rho$ symbolizes the density, $t$ denotes the time, $f_i$ denotes the body force, and
$\lambda$ alongside $\mu$ represents the Lamé coefficients. In the nonlinear small deformation
scenario, substituting the Green strain tensor and its corresponding volumetric strain
into constitutive equation and equations of motion, culminating in the formulation of





the subsequent equation:

$$
\rho \frac{\partial^2 u_i}{\partial t^2} = \rho f_i + (\lambda + \mu)\frac{\partial \theta}{\partial x_i} + \mu \frac{\partial^2 u_i}{\partial x_j \partial x_j}
$$
$$
+ \lambda \left[ \frac{\partial^2 u_k}{\partial x_i x_j}\cdot\frac{\partial u_k}{\partial x_j} + \frac{\partial}{\partial x_i}\left( \frac{\partial u_x}{\partial x_x}\cdot\frac{\partial u_y}{\partial x_y} + \frac{\partial u_x}{\partial x_x}\cdot\frac{\partial u_z}{\partial x_z} + \frac{\partial u_y}{\partial x_y}\cdot\frac{\partial u_z}{\partial x_z} \right) \right] + \mu\left( \frac{\partial^2 u_k}{\partial x_i x_j}\cdot\frac{\partial u_k}{\partial x_j} + \frac{\partial^2 u_k}{\partial x_j x_j}\cdot\frac{\partial u_k}{\partial x_i} \right)
$$

135                                                                                                                    (10)

Eq. (10) introduces several third-order terms diverging from the composition of
Eq. (9). Their difference unveils the nonlinearity in terms of the displacement field **u**
and the elastic parameters ($\mu$ and $\lambda$) under nonlinear small deformation. Eq. (10)
exhibits more complexity, signifying the introduction of additional physical intricacies
into an elastomer's deformation dynamics. The inclusion of nonlinear terms describes
the nonlinear response of the medium by linking it with the shear modulus ($\mu$) and the
bulk modulus ($\lambda$), thereby impacting the propagation attributes of elastic waves. The
increment of the equation associated with the shear modulus $\mu$ engenders nonlinear
effects via the strain tensor, while the increment associated with the bulk modulus $\lambda$
induces nonlinear effects through the volumetric strain.
The disparity between the two wave equations does not directly translate to the
final displacement field discrepancies. The displacement field in Eq. (10) is the result
of the nonlinear small deformation, in contrast to Eq. (9), where such nonlinear effects
are absent. Therefore, the velocity-stress equations using the Green strain tensor are
derived next to compare the difference in wave fields between the two by numerical
simulation of seismic wavefields.

2.2 Velocity-stress elastic wave equations
The staggered-grid finite-difference method is well-established for performing





numerical simulations of seismic wavefields. By discretizing the medium and the
wave equations, the numerical solution of the wavefield is obtained at each grid point
under each time node as time progresses. In general, the first-order velocity-stress
elastic wave equations under the assumption of linear small deformation in
3-dimensional (3D) isotropic media are

$$
\begin{cases}
\dfrac{\partial \sigma_{xx}}{\partial x}+\dfrac{\partial \sigma_{xy}}{\partial y}+\dfrac{\partial \sigma_{xz}}{\partial z}+f_x=\rho\dfrac{\partial v_x}{\partial t}\\[2mm]
\dfrac{\partial \sigma_{yx}}{\partial x}+\dfrac{\partial \sigma_{yy}}{\partial y}+\dfrac{\partial \sigma_{yz}}{\partial z}+f_y=\rho\dfrac{\partial v_y}{\partial t}\\[2mm]
\dfrac{\partial \sigma_{zx}}{\partial x}+\dfrac{\partial \sigma_{zy}}{\partial y}+\dfrac{\partial \sigma_{zz}}{\partial z}+f_z=\rho\dfrac{\partial v_z}{\partial t}
\end{cases}
$$

$$
\begin{cases}
\dfrac{\partial \sigma_{xx}}{\partial t}=(\lambda+2\mu)\dfrac{\partial v_x}{\partial x}+\lambda\dfrac{\partial v_y}{\partial y}+\lambda\dfrac{\partial v_z}{\partial z}\\[2mm]
\dfrac{\partial \sigma_{yy}}{\partial t}=\lambda\dfrac{\partial v_x}{\partial x}+(\lambda+2\mu)\dfrac{\partial v_y}{\partial y}+\lambda\dfrac{\partial v_z}{\partial z}\\[2mm]
\dfrac{\partial \sigma_{zz}}{\partial t}=\lambda\dfrac{\partial v_x}{\partial x}+\lambda\dfrac{\partial v_y}{\partial y}+(\lambda+2\mu)\dfrac{\partial v_z}{\partial z}\\[2mm]
\dfrac{\partial \sigma_{xy}}{\partial t}=\mu(\dfrac{\partial v_y}{\partial x}+\dfrac{\partial v_x}{\partial y})\\[2mm]
\dfrac{\partial \sigma_{xz}}{\partial t}=\mu(\dfrac{\partial v_z}{\partial x}+\dfrac{\partial v_x}{\partial z})\\[2mm]
\dfrac{\partial \sigma_{yz}}{\partial t}=\mu(\dfrac{\partial v_z}{\partial y}+\dfrac{\partial v_y}{\partial z})
\end{cases} \tag{11}
$$

$$
\begin{cases}
R_x=\dfrac{1}{2}(\dfrac{\partial v_z}{\partial y}-\dfrac{\partial v_y}{\partial z})\\[2mm]
R_y=\dfrac{1}{2}(\dfrac{\partial v_x}{\partial z}-\dfrac{\partial v_z}{\partial x})\\[2mm]
R_z=\dfrac{1}{2}(\dfrac{\partial v_y}{\partial x}-\dfrac{\partial v_x}{\partial y})
\end{cases}
$$

Where $\sigma_{ji}$ denotes the stress tensor, $v_x$, $v_y$, and $v_z$ denote the velocity of X, Y, and Z
components. $R_{xz}$ corresponds to the rotation rate around Y axis, commonly referred to
as $R_Y$ in rotational seismology, as well as $R_X$ and $R_Z$.

Similarly, the velocity-stress elastic wave equations under the assumption of

nonlinear small deformation in 3D isotropic media can be given as Eq. (12).






$$\begin{cases} \dfrac{\partial \sigma_{xx}}{\partial x} + \dfrac{\partial \sigma_{xy}}{\partial y} + \dfrac{\partial \sigma_{xz}}{\partial z} + f_x = \rho \dfrac{\partial v_x}{\partial t} \\[2mm] \dfrac{\partial \sigma_{yx}}{\partial x} + \dfrac{\partial \sigma_{yy}}{\partial y} + \dfrac{\partial \sigma_{yz}}{\partial z} + f_y = \rho \dfrac{\partial v_y}{\partial t} \\[2mm] \dfrac{\partial \sigma_{zx}}{\partial x} + \dfrac{\partial \sigma_{zy}}{\partial y} + \dfrac{\partial \sigma_{zz}}{\partial z} + f_z = \rho \dfrac{\partial v_z}{\partial t} \end{cases}$$

$$\begin{cases} \dfrac{\partial \sigma_{xx}}{\partial t} = (\lambda+2\mu)\dfrac{\partial v_x}{\partial x} + \lambda\dfrac{\partial v_y}{\partial y} + \lambda\dfrac{\partial v_z}{\partial z} + 2\,dt\cdot\lambda\cdot\left(\dfrac{\partial v_x}{\partial x}\dfrac{\partial v_y}{\partial y} + \dfrac{\partial v_x}{\partial x}\dfrac{\partial v_z}{\partial z} + \dfrac{\partial v_y}{\partial y}\dfrac{\partial v_z}{\partial z}\right) + dt\cdot(\lambda+2\mu)\cdot\left(\dfrac{\partial v_x}{\partial x}\dfrac{\partial v_x}{\partial x} + \dfrac{\partial v_y}{\partial x}\dfrac{\partial v_y}{\partial x} + \dfrac{\partial v_z}{\partial x}\dfrac{\partial v_z}{\partial x}\right) \\[2mm] \quad + dt\cdot\lambda\cdot\left(\dfrac{\partial v_x}{\partial y}\dfrac{\partial v_x}{\partial y} + \dfrac{\partial v_y}{\partial y}\dfrac{\partial v_y}{\partial y} + \dfrac{\partial v_z}{\partial y}\dfrac{\partial v_z}{\partial y}\right) + dt\cdot\lambda\cdot\left(\dfrac{\partial v_x}{\partial z}\dfrac{\partial v_x}{\partial z} + \dfrac{\partial v_y}{\partial z}\dfrac{\partial v_y}{\partial z} + \dfrac{\partial v_z}{\partial z}\dfrac{\partial v_z}{\partial z}\right) \\[2mm] \dfrac{\partial \sigma_{yy}}{\partial t} = \lambda\dfrac{\partial v_x}{\partial x} + (\lambda+2\mu)\dfrac{\partial v_y}{\partial y} + \lambda\dfrac{\partial v_z}{\partial z} + 2\,dt\cdot\lambda\cdot\left(\dfrac{\partial v_x}{\partial x}\dfrac{\partial v_y}{\partial y} + \dfrac{\partial v_x}{\partial x}\dfrac{\partial v_z}{\partial z} + \dfrac{\partial v_y}{\partial y}\dfrac{\partial v_z}{\partial z}\right) + dt\cdot\lambda\cdot\left(\dfrac{\partial v_x}{\partial x}\dfrac{\partial v_x}{\partial x} + \dfrac{\partial v_y}{\partial x}\dfrac{\partial v_y}{\partial x} + \dfrac{\partial v_z}{\partial x}\dfrac{\partial v_z}{\partial x}\right) \\[2mm] \quad + dt\cdot(\lambda+2\mu)\cdot\left(\dfrac{\partial v_x}{\partial y}\dfrac{\partial v_x}{\partial y} + \dfrac{\partial v_y}{\partial y}\dfrac{\partial v_y}{\partial y} + \dfrac{\partial v_z}{\partial y}\dfrac{\partial v_z}{\partial y}\right) + dt\cdot\lambda\cdot\left(\dfrac{\partial v_x}{\partial z}\dfrac{\partial v_x}{\partial z} + \dfrac{\partial v_y}{\partial z}\dfrac{\partial v_y}{\partial z} + \dfrac{\partial v_z}{\partial z}\dfrac{\partial v_z}{\partial z}\right) \\[2mm] \dfrac{\partial \sigma_{zz}}{\partial t} = \lambda\dfrac{\partial v_x}{\partial x} + \lambda\dfrac{\partial v_y}{\partial y} + (\lambda+2\mu)\dfrac{\partial v_z}{\partial z} + 2\,dt\cdot\lambda\cdot\left(\dfrac{\partial v_x}{\partial x}\dfrac{\partial v_y}{\partial y} + \dfrac{\partial v_x}{\partial x}\dfrac{\partial v_z}{\partial z} + \dfrac{\partial v_y}{\partial y}\dfrac{\partial v_z}{\partial z}\right) + dt\cdot\lambda\cdot\left(\dfrac{\partial v_x}{\partial x}\dfrac{\partial v_x}{\partial x} + \dfrac{\partial v_y}{\partial x}\dfrac{\partial v_y}{\partial x} + \dfrac{\partial v_z}{\partial x}\dfrac{\partial v_z}{\partial x}\right) \\[2mm] \quad + dt\cdot\lambda\cdot\left(\dfrac{\partial v_x}{\partial y}\dfrac{\partial v_x}{\partial y} + \dfrac{\partial v_y}{\partial y}\dfrac{\partial v_y}{\partial y} + \dfrac{\partial v_z}{\partial y}\dfrac{\partial v_z}{\partial y}\right) + dt\cdot(\lambda+2\mu)\cdot\left(\dfrac{\partial v_x}{\partial z}\dfrac{\partial v_x}{\partial z} + \dfrac{\partial v_y}{\partial z}\dfrac{\partial v_y}{\partial z} + \dfrac{\partial v_z}{\partial z}\dfrac{\partial v_z}{\partial z}\right) \\[2mm] \dfrac{\partial \sigma_{xy}}{\partial t} = \mu\left(\dfrac{\partial v_y}{\partial x} + \dfrac{\partial v_x}{\partial y}\right) + dt\cdot2\mu\cdot\left(\dfrac{\partial v_x}{\partial x}\dfrac{\partial v_x}{\partial y} + \dfrac{\partial v_y}{\partial x}\dfrac{\partial v_y}{\partial y} + \dfrac{\partial v_z}{\partial x}\dfrac{\partial v_z}{\partial y}\right) \\[2mm] \dfrac{\partial \sigma_{xz}}{\partial t} = \mu\left(\dfrac{\partial v_x}{\partial z} + \dfrac{\partial v_z}{\partial x}\right) + dt\cdot2\mu\cdot\left(\dfrac{\partial v_x}{\partial z}\dfrac{\partial v_x}{\partial x} + \dfrac{\partial v_y}{\partial z}\dfrac{\partial v_y}{\partial x} + \dfrac{\partial v_z}{\partial z}\dfrac{\partial v_z}{\partial x}\right) \\[2mm] \dfrac{\partial \sigma_{yz}}{\partial t} = \mu\left(\dfrac{\partial v_z}{\partial y} + \dfrac{\partial v_y}{\partial z}\right) + dt\cdot2\mu\cdot\left(\dfrac{\partial v_x}{\partial y}\dfrac{\partial v_x}{\partial z} + \dfrac{\partial v_y}{\partial y}\dfrac{\partial v_y}{\partial z} + \dfrac{\partial v_z}{\partial y}\dfrac{\partial v_z}{\partial z}\right) \end{cases} \quad (12)$$

$$\begin{cases} R_x = \dfrac{1}{2}\left(\dfrac{\partial v_z}{\partial y} - \dfrac{\partial v_y}{\partial z}\right) \\[2mm] R_y = \dfrac{1}{2}\left(\dfrac{\partial v_x}{\partial z} - \dfrac{\partial v_z}{\partial x}\right) \\[2mm] R_z = \dfrac{1}{2}\left(\dfrac{\partial v_y}{\partial x} - \dfrac{\partial v_x}{\partial y}\right) \end{cases}$$

Where the variables and symbols are defined in the same way as in Eq. (11).

2.3 Staggered-grid finite difference method
This study utilizes the staggered-grid finite difference method to simulate the
seismic wavefields (Sun et al., 2018). The model is divided into two sets of grids,
wherein the velocity and stress of the medium are defined in separate grid systems
(Madariaga, 1976). The grid configuration for a two-dimensional model scenario is
illustrated in Fig. 2.



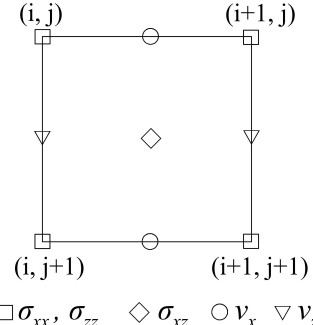


**Figure 2.** Schematic diagram of 2D staggered grids.

Based on Eqs. (11) and (12), we can simulate the seismic waves propagating in
discrete grids with two-order time and six-order space differential approximations. To
weaken the boundary reflections, perfectly matched absorbing layer boundary
conditions are adapted to the boundaries (Dong & Ma 2000). Alternatively, the
acoustic boundary replacement method is adopted to ensure the free-surface condition
at the upper boundary (Xu et al., 2007; Wang et al., 2012).

2.4 Simulation parameters
For the physical process of source excitation, when the seismic wavelength under
study substantially surpasses the scale of the involved source, the seismic source can
be regarded as a point source. The seismic moment tensor, represented by equation
(13), is the most comprehensive depiction of seismic point sources.
$$\boldsymbol{M} = \begin{pmatrix} M_{xx} & M_{xy} & M_{xz} \\ M_{yx} & M_{yy} & M_{yz} \\ M_{zx} & M_{zy} & M_{zz} \end{pmatrix}, i, j = x, y, z \tag{13}$$

In Eq. (11), $M_{ij}$ represents each moment-element component. The first index
signifies the force direction, and the second index signifies the direction of force arm.



The moment tensor can be decomposed into three distinct parts: the isotropy
component (ISO), the double couple component (DC), and the compensated linear
vector dipole component (CLVD) (Knopoff and Randall, 1970). The ISO component
represents the volume expansion of the focal area, characterized by a non-zero trace
and uniform force and direction of force arm in three vector dipoles. The DC
component denotes the dislocation of the two fault walls without volume variation.
The CLVD component is also composed of three vector dipoles, with one being twice
as large as the other two. The three basic seismic source components can be expressed
as follows.
$$\boldsymbol{M}^{ISO} = \begin{pmatrix} M_{xx} & 0 & 0 \\ 0 & M_{yy} & 0 \\ 0 & 0 & M_{zz} \end{pmatrix} \tag{14}$$

$$\boldsymbol{M}^{DC} = \begin{pmatrix} 0 & M_{xy} & 0 \\ M_{yx} & 0 & 0 \\ 0 & 0 & 0 \end{pmatrix} \tag{15}$$

$$\boldsymbol{M}^{CLVD} = \begin{pmatrix} M_{xx} & 0 & 0 \\ 0 & M_{yy} & 0 \\ 0 & 0 & -2M_{zz} \end{pmatrix} \tag{16}$$


In numerical simulations, the Ricker wavelet with a central frequency of 0.5 Hz is
employed as a seismic source wavelet to simulate the three simplest sources: ISO, DC,
and CLVD. The body force, represented by the moment tensor, can be converted into
a velocity source by incrementally being added to individual velocity components to
simulate the three basic sources (Graves, 1996). The specific loading equations in the
grid system are outlined below.



$$\text{ISO}: \begin{cases} \Delta v_x^{n+\frac{1}{2}}\left(i+\frac{1}{2},j,k\right)=\dfrac{M_{xx}dt}{\rho Vdx}f^n \\[1mm] \Delta v_x^{n+\frac{1}{2}}\left(i-\frac{1}{2},j,k\right)=\dfrac{-M_{xx}dt}{\rho Vdx}f^n \\[1mm] \Delta v_y^{n+\frac{1}{2}}\left(i,j+\frac{1}{2},k\right)=\dfrac{M_{yy}dt}{\rho Vdy}f^n \\[1mm] \Delta v_y^{n+\frac{1}{2}}\left(i,j-\frac{1}{2},k\right)=\dfrac{-M_{yy}dt}{\rho Vdy}f^n \\[1mm] \Delta v_z^{n+\frac{1}{2}}\left(i,j,k+\frac{1}{2}\right)=\dfrac{M_{zz}dt}{\rho Vdz}f^n \\[1mm] \Delta v_z^{n+\frac{1}{2}}\left(i,j,k-\frac{1}{2}\right)=\dfrac{-M_{zz}dt}{\rho Vdz}f^n \end{cases} \tag{17}$$


$$\text{DC}: \begin{cases} \Delta v_x^{n+\frac{1}{2}}\left(i+\frac{1}{2},j,k\right)=\dfrac{-M_{xy}dt}{\rho Vdy}f^n \\[1mm] \Delta v_x^{n+\frac{1}{2}}\left(i+\frac{1}{2},j-1,k\right)=\dfrac{M_{xy}dt}{\rho Vdy}f^n \\[1mm] \Delta v_y^{n+\frac{1}{2}}\left(i,j-\frac{1}{2},k\right)=\dfrac{M_{yx}dt}{\rho Vdx}f^n \\[1mm] \Delta v_y^{n+\frac{1}{2}}\left(i+1,j-\frac{1}{2},k\right)=\dfrac{-M_{yx}dt}{\rho Vdx}f^n \end{cases} \tag{18}$$


$$\text{CLVD}: \begin{cases} \Delta v_x^{n+\frac{1}{2}}\left(i+\frac{1}{2},j,k\right)=\dfrac{M_{xx}dt}{\rho Vdx}f^n \\[1mm] \Delta v_x^{n+\frac{1}{2}}\left(i-\frac{1}{2},j,k\right)=\dfrac{-M_{xx}dt}{\rho Vdx}f^n \\[1mm] \Delta v_y^{n+\frac{1}{2}}\left(i,j+\frac{1}{2},k\right)=\dfrac{M_{yy}dt}{\rho Vdy}f^n \\[1mm] \Delta v_y^{n+\frac{1}{2}}\left(i,j-\frac{1}{2},k\right)=\dfrac{-M_{yy}dt}{\rho Vdy}f^n \\[1mm] \Delta v_z^{n+\frac{1}{2}}\left(i,j,k+\frac{1}{2}\right)=\dfrac{-2M_{zz}dt}{\rho Vdz}f^n \\[1mm] \Delta v_z^{n+\frac{1}{2}}\left(i,j,k-\frac{1}{2}\right)=\dfrac{2M_{zz}dt}{\rho Vdz}f^n \end{cases} \tag{19}$$

$\Delta v$ denotes the velocity increment; $n$ denotes the time sampling node; $dt$, $\rho$, and $V$
represent the time sampling interval, medium density, and the medium model's unit
volume, respectively. The source-time function $f^n$ denotes the Ricker wavelet's
amplitude at the corresponding time node.



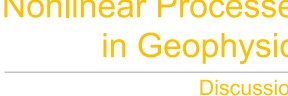
To focus on the influence of different small deformation scenarios on seismic
elastic waves, we only discuss the characteristics in a 3D isotropic full-space
homogeneous medium. The model is set with a size of 60 km ($x$) × 60 km ($y$)× 60 km
($z$), with mesh division spacing set at 0.5 km. Model properties include $v_p$=4400 m/s,
$v_s$=3000 m/s, and $\rho$=2600 kg/m$^3$. The moment source is positioned at the model's
center, where x=y=z=30 km. The time sampling interval is 15 ms, and the total
recording time spans 10 seconds.

**3 Wavefield simulations of three types of basic seismic source**
3.1 ISO source
Under the assumption of nonlinear small deformation related to the condition of
the Green strain tensor, the 3-component translational and rotational seismic
snapshots are synthesized and illustrated in Fig. 3a. These snapshots demonstrate the
generation of solely P-wave, with minimal energy projected in rotational components
upon the excitation of ISO source.
To highlight the distinction in wave propagation between linear and nonlinear
conditions, we present the wavefield difference and their approximation with the
relative change in Fig. 3b and c. Minimal disparities are observed in P-wave fronts,
indicating that the assumption of linear small deformation is satisfied for P-wave in
ISO source simulation. Conversely, examining the S-wave fronts in Fig. 3b and their
relative changes (ranging approximately between 5-20 percent) in Fig. 3c lead to the



conclusion that even in the ISO simulation, the coupling of P- and S-waves in the
wave equations allows the generation of S-waves, a phenomenon that is unattainable
under conditions of linear small deformation.

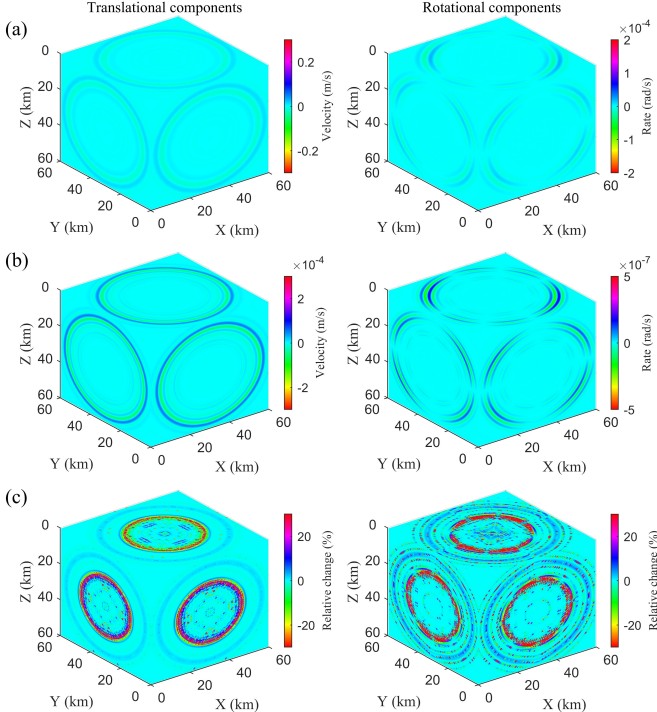


**Figure 3.** Wavefield comparisons at 8th second excited by ISO source. (a) presents

the wavefield snapshots under nonlinear small deformation, (b) presents the

difference between linear and nonlinear conditions, and (c) presents their relative

change in percentage (using the linear result as the denominator)


3.2 DC source

The wavefields excited by the DC source are illustrated in Fig. 4a, revealing the

generation of relatively weak P and stronger S waves. The application of double force

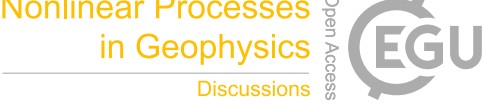

moments ($M_{xy}$ and $M_{yx}$) loaded within the x-y plane results in the X- and
Y-components of translational motions being stronger than the Z-component.
Consequently, the $\mathbf{R}_Z$ exhibits a greater degree of wavefield energy than the $\mathbf{R}_X$ and
$\mathbf{R}_Y$ components. From the wavefield differences and relative change between the two
assumptions (Fig. 4b and c), it becomes evident that the discrepancy in S-wave is
notable, and the relative change in P wave is more prominent in the rotational
components (below 10 %). Moreover, the distinction in the wavefront polarity of the
P- and S-wave in the wavefield caused by nonlinearity is totally different from the
polarity of the wavefield itself, as illustrated in Fig. 4a.

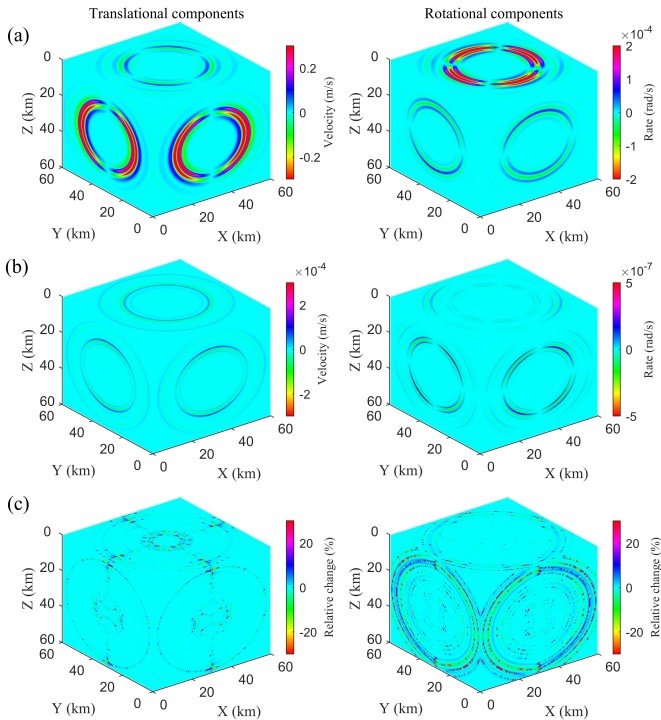


**Figure 4.** Wavefield comparisons at 8th second excited by DC source. (a)
presents the wavefield snapshots under nonlinear small deformation, (b) presents the





difference between the linear and nonlinear conditions, and (c) presents their relative

change with percentage (using the linear result as the denominator)


3.3 CLVD source

Fig. 5a displays the results generated by CLVD source. In comparison to the

outcomes of ISO and DC sources, the CLVD elicits more pronounced S waves
primarily projected in $\mathbf{R}_X$ and $\mathbf{R}_Y$ components. Moreover, the wavefield differences
between linearity and nonlinearity intensify, particularly in S wave in rotational
motion (Fig. 5b). Their maximum relative change can reach up to 10 percent,
especially along the diagonal direction of 45 degrees (Fig. 5c).

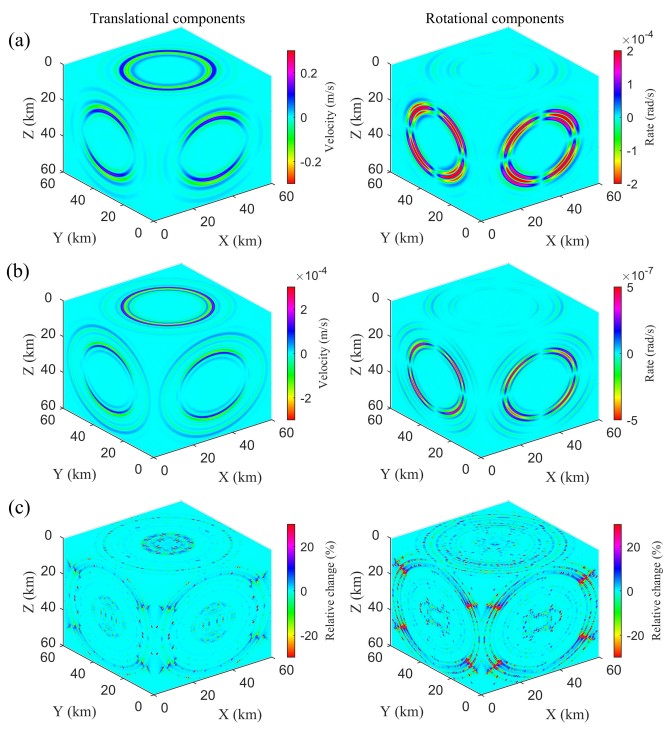


**Figure 5.** Wavefield comparisons at 8th second excited by CLVD source. (a)



presents the wavefield snapshots under nonlinear small deformation, (b) presents the
difference between the linear and nonlinear conditions, and (c) presents their relative
change with percentage (using the linear result as the denominator)

3.4 Comparisons of wavefield energy for basic seismic sources
The disparities in propagation of nonlinear elastic waves in homogeneous media
are predominantly observed in rotational components, as evidenced by the
aforementioned comparisons and analyses. Further calculating the wavefield energy
for the above wavefield snapshot display area and comparing the variations of wave
energy in relative changes over time progression and the change at the 8th second
with the seismic moment magnitude increasing, as illustrated in Fig. 6. In Fig. 6a, the
overall errors in wavefield energy consistently remain below 1 percent as the wave
propagates near the source area with small magnitude, signifying that the linear
assumption is adequate for the three basic moment tensor sources. In Fig. 6b, the
changing curves for the DC source display less smoothness than those for the CLVD,
and the relative change in rotational components consistently outweighs this in
translational components. Moreover, the curves demonstrate a nearly exponential
increase with rising earthquake magnitude. Upon reaching a strong magnitude of 7,
especially for the ISO source, the errors in rotational motions reach 25 percent, while
these in translation amount to approximately 10 percent. The error due to CLVD
sources can also reach about 5 %, while the DC-induced error remains small. Because
the DC source component typically dominates the focal mechanisms for the majority



of earthquakes, as opposed to the ISO component (Zhao and Zhang, 2022), it can be
inferred that the approximation of linear scenario is well-suited for the majority of
seismic body waves simulations, except in instances of strong seismic activity.

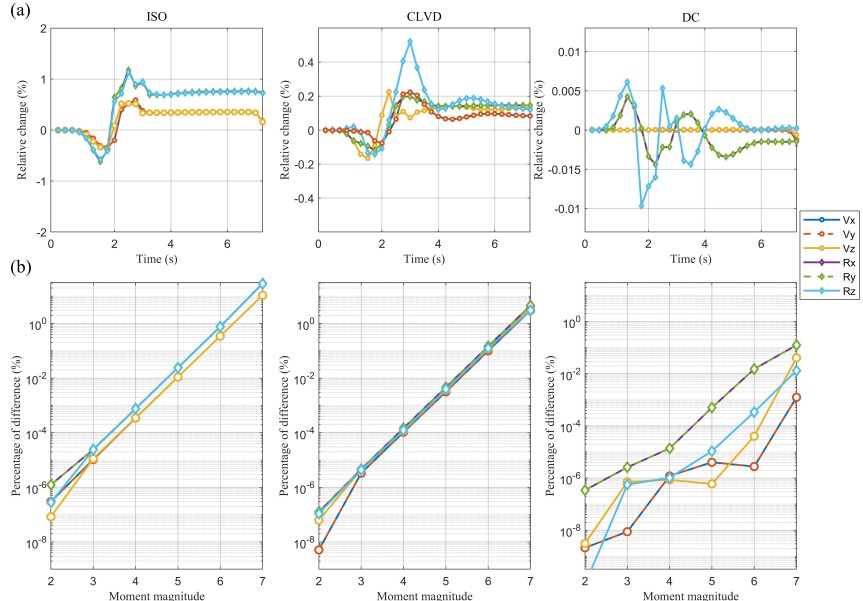


**Figure 6.** Relative changes of wavefield energy induced by nonlinearity with (a)
spreading time and (b) increasing earthquake magnitude

## 4 Seismic observations and simulations of two Taiwan earthquakes

4.1 Hualien earthquakes
Taiwan, situated at the confluence of three significant tectonic plates - the
Philippine Sea Plate, the Eurasia Plate, and the Pacific Ocean Plate, experiences
frequent seismic activity, particularly moderate to large earthquakes annually (Zheng
et al., 2005). The 2018 Hualien earthquake with a magnitude of $M_W$ 5.41 (referred to



as **E**1) and the 2019 Hualien earthquake with a magnitude of $M_W$ 6.13 (referred to as
**E**2), with epicenter depths of 15 km and 30 km, respectively, occurred off the eastern
coast of Taiwan. The epicenter locations and station placements depicted by GMT are
shown in Fig. 7 (Wessel et al., 2019). The receiver for **E**2, located in Fujian province,
is positioned 327 km from the epicenter (Fig. 7a). Additionally, a seismic array
comprising seven 3C translational seismometers was deployed approximately 53 km
from the epicenter of **E**1    (Yuan et al., 2020) (Fig. 7b). A blueSeis-3A fiber-optic
rotational seismometer was placed at the NA01 station in the center of the array to
directly record the seismic rotational rates (Bernauer et al. 2018; Cao et al., 2021).
According to the monitoring data from the U.S. Geological Survey (USGS,
https://www.usgs.gov/), both **E**1 and **E**2 were triggered by reverse faults, and beach
balls representing their focal mechanisms are shown in Fig. 7c. The moment tensor
parameters of **E**1 and **E**2 are presented in Eqs. (20) and (21), respectively.



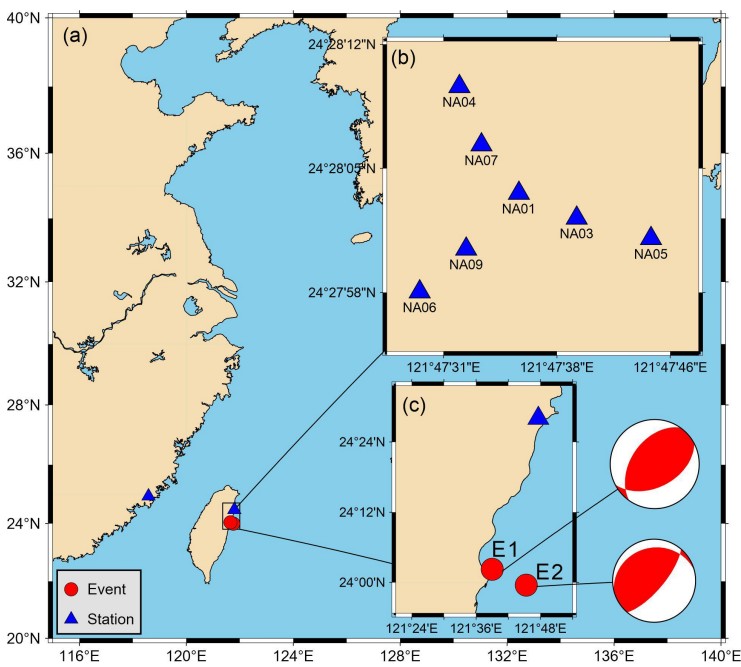

**Figure 7.** Epicenters and observation sites of the two earthquakes

$$M_{rr} = 9.942 \times 10^{16}, M_{tt} = -7.569 \times 10^{16}, M_{pp} = -2.373 \times 10^{16}$$
$$M_{rt} = 7.372 \times 10^{16}, M_{rp} = 1.0965 \times 10^{17}, M_{tp} = -4.156 \times 10^{16}$$
(20)

$$M_{rr} = 1.8247 \times 10^{18}, M_{tt} = -1.064 \times 10^{18}, M_{pp} = -7.607 \times 10^{17}$$
$$M_{rt} = 3.141 \times 10^{17}, M_{rp} = -3.155 \times 10^{17}, M_{tp} = -1.114 \times 10^{18}$$
(21)

4.2 Wavefield simulations of the Taiwan earthquakes

To simulate **E**1 and **E**2, we implement the free-surface condition at the upper

surface and absorbing boundary conditions in other directions of the 3D model.

According to the CRUST1.0 model (Laske et al., 2013), the subsurface medium at the

**E**1 observation station is divided into five distinct layers, as detailed in Table 1. The

3D model is constructed with a size of 60 km ($x$, NS) × 20 km ($y$, EW) × 30 km ($z$,

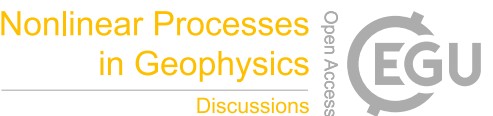

vertical) to suit the specifics of the observation system, with the corresponding
parameters shown in Table 2.
**Table 1** Underground layered medium at observing stations

| Layer | Thickness (km) | Vp (km/s) | Vs (km/s) | $\rho$ (kg/m³) |
|-------|----------------|-----------|-----------|----------------|
| 1 | 0.50 | 2.50 | 1.07 | 2.11 |
| 2 | 10.12 | 5.80 | 3.40 | 2.63 |
| 3 | 9.81 | 6.30 | 3.62 | 2.74 |
| 4 | 9.82 | 6.90 | 3.94 | 2.92 |
| 5 | - | 7.70 | 4.29 | 3.17 |


**Table 2** Parameters for simulating model 1 (**E**1)

| Items | Parameters |
|-------|------------|
| Source type | Eq. (20) |
| Central frequency | 1 Hz |
| Grid interval | 1 km |
| Time interval | 5 ms |
| Source position | (0, 0, 15 km) |
| Receiver position | (53 km, : , 0 km) |
| Recording time | 30 s |


Sorting the synthetic records from model 1 at coordinates X=53 km and Y=4 km,
corresponding to the NA01 station, the seismic waveforms are presented in Fig. 8a. It
can be found that, apart from direct P- and S-waves, **E**1 predominantly exhibits
elliptical polarization in X-Z vertical plane and rotational movements around Y-axis
induced by Rayleigh wave in the north-south vertical plane. The large order of
magnitude difference in amplitude between theoretical simulations and actual
observations is due to the assumption of elastic media, though the actual propagation
media are usually viscoelastic, which will absorb and attenuate seismic energy and
high frequency.



The unavoidable site effect leads to the practical observation in Fig. 8b displaying
significantly stronger horizontal components than vertical ones (Abercrombie, 1997;
Guatteri et al., 2001). The site effect and the nearly northeast strike of the seismogenic
fault result in pronounced translational components and $\mathbf{R}_Z$ component recordings
mixed with complex seismic waves after P- and S-wave arrivals, indicating the
presence of Love waves and significant disparities between the actual Earth's medium
and the simplified Crust model. Fig. 8 also shows that the simulated rotational
components are 1000 times of magnitude weaker than the simulated translational
components, but the observed rotational motions are 250 times weaker than the
translational ones.

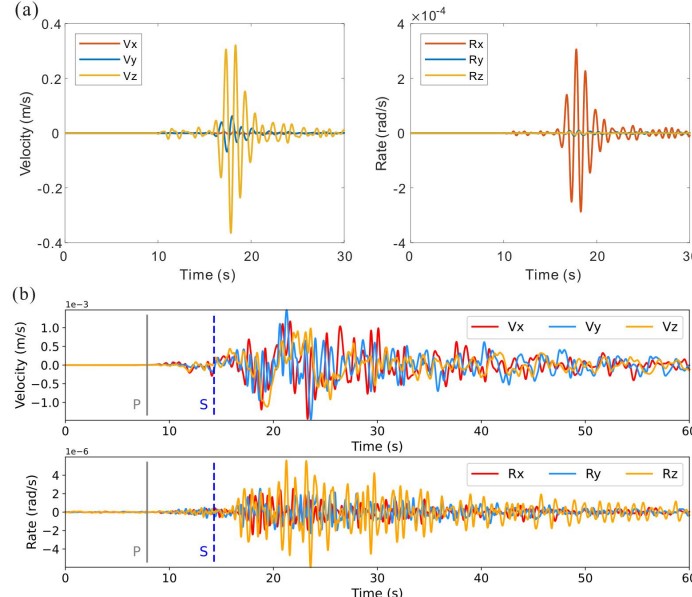


**Figure 8.** 6C seismic records of (a) theoretical simulation under linear small
deformation for $\mathbf{E}$1. In (b), for the real seismic records, a band-pass filter of 0.1 Hz to
2 Hz is applied, and the corresponding arrival times of P and S waves are calculated



according to the iasp91 model (Kennett and Engdahl, 1991)

**Table 3** Parameters for simulating model 2 (**E**2).

| Items | Parameters |
|---|---|
| Source type | Eq. (21) |
| Central frequency | 0.5 Hz |
| Grid interval | 5 km |
| Time interval | 2 ms |
| Source position | (0, 310 km, 30 km) |
| Receiver position | (:, 0 km, 0 km) |
| Recording time | 300 s |


The same modeling approach is adopted to simulate **E**2, with the parameters of
model 2 detailed in Table 3, featuring a size of 150 km (x, NS) × 350 km (y, EW) × 50
km (z, vertical). The 6C seismic recordings at X=100 km and Y=0 km, corresponding
to the receiver station, are extracted from the simulation result of model 2, as
displayed in Fig. 9a. The simulated records show a dominance of $\mathbf{V}_Z$ over $\mathbf{V}_X$ and $\mathbf{V}_Y$
components, with $\mathbf{R}_X$ and $\mathbf{R}_Y$ components exhibiting more strength than $\mathbf{R}_Z$
component, showcasing the rotational motions primarily occurring in the horizontal
direction. In addition to the direct P and S waves and surface waves, this intense
seismic shock generated strong secondary waves. In the actual observation records
(Fig. 9b), where the station is located on solid rock within a tunnel, the $\mathbf{V}_Z$ component
is slightly stronger than the $\mathbf{V}_X$ and $\mathbf{V}_Y$ components, while the $\mathbf{R}_Z$ component is
slightly weaker than the $\mathbf{R}_X$ and $\mathbf{R}_Y$ components. This suggests that the rotational
motions for **E**2 are predominantly in horizontal directions, and the site effect is
relatively weaker. In addition, the amplitude difference between the actual observed





rotational and translational components is smaller than the amplitude difference
between the simulated advective and rotational components, and the observed
rotational component is relatively stronger, which is the same as the characteristic
shown in Fig. 9. That is consistent with previous studies that have argued that the
observed rotational components have a relatively stronger amplitude than the
rotational component converted from translational components (Teisseyre et al.,

2003).

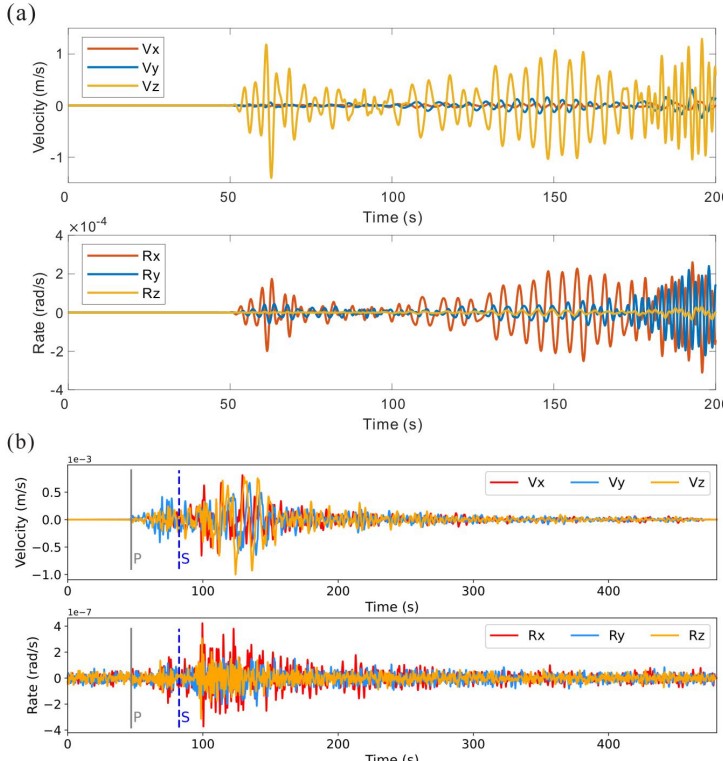


**Figure 9.** 6C seismic records of (a) theoretical simulation under linear small
deformation for **E2**. In (b), for the real seismic records, a band-pass filter of 0.1 Hz to
1 Hz is applied, and the corresponding arrival times of P and S waves are calculated



according to the iasp91 model (Kennett and Engdahl, 1991)

Following the numerical simulation of $\mathbf{E}1$ and $\mathbf{E}2$ under the conditions of linear

and nonlinear simulations, respectively, we make a theoretical comparison by

calculating the relative differences between the two scenarios. The relative changes in

root-mean-square (RMS) amplitude are used to compare the linear errors of these two

earthquakes. The RMS amplitude values of the waveforms recorded in a 2-s time

window are calculated at 1-s intervals to reflect the energy of the seismic recordings,

and then the relative change percentage of RMS amplitude of the nonlinear simulation

results relative to that of the linear simulation is derived accordingly, and the results

are shown in Fig. 10.

In Fig. 10a, it can be seen that the error of the nonlinear simulation of $\mathbf{E}1$ is very

small relative to the linear simulation, and only the error on the $\mathbf{V}_X$ component is

slightly larger but is less than 0.4 %. This indicates that for the simulation of $\mathbf{E}1$, the

error introduced by the linear approximation is basically negligible. For the results of

$\mathbf{E}2$ in Fig. 10b, the translational components show larger errors than the rotational

components, especially the $\mathbf{V}_X$ and $\mathbf{V}_Y$ components, with errors up to 10 %, and the

errors on the $\mathbf{V}_Z$ components are basically within 5 %; the linear approximation errors

on the three rotational components are even smaller, basically within 2 %. For the

body waves dominated records before 120 s, $\mathbf{R}_X$ and $\mathbf{R}_Y$ components reflect a larger

error percentage than $\mathbf{R}_Z$ component. In the surface-wave records after the 150 s, the

$\mathbf{R}_Z$ component shows increased nonlinear errors. These results indicate that the linear

simplification of rotation for the elastomer strain process has a small error for the





rotational component but produces a larger wavefield error on the translational
components.
The linear approximation produces more errors on the translational components
obtained from real earthquake simulations, probably because the wavefield energy of
rotational component decays faster in natural earthquakes (Lee et al., 2009; Lai and
Sun, 2017). Besides, the simulation results of **E**2 show a larger difference between
linearity and nonlinearity than that of **E**1, which is about ten times larger, mainly
because of the increased source energy of **E**2. So, for weak and moderate earthquakes,
the effect of nonlinearity may be negligible, and the linear approximation can meet
the research accuracy. It can also be attributed to the fact that the two earthquakes
have different source mechanisms, which makes its linear approximation error larger.

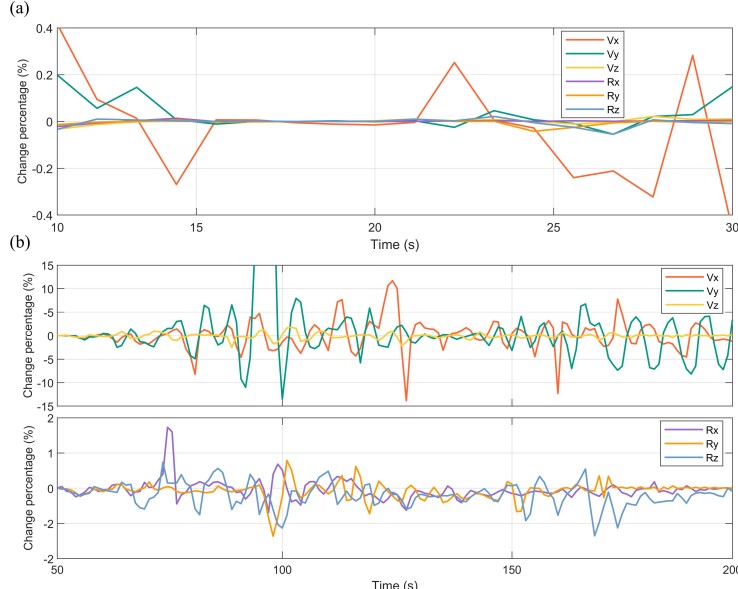


**Figure 10.** Relative changes in RMS amplitude of simulation results between linear
and nonlinear scenarios for **E**1 (a) and **E**2 (b)




**5 Discussions**

Compared with the traditional theory of seismic wave propagation in
homogeneous elastic media, the Green strain tensor is a function of both the strain
tensor and the rotation tensor, as shown in Eq. (5). Without considering the linear
approximation of small deformation, the wave propagation equations entail
three-order differentiations of displacement, with the higher-order terms influenced by
shear modulus and bulk modulus. Given that earthquakes mostly occur in shallow
crust or transitional zones between shell and mantle, often considered as planes of
elastic attributes transformation and stress discontinuity zones, more intricate media
and focal physics (Olson and Apsel, 1982; Olson and Allen, 2005), such as the model
featuring a rigid thin-layer sphere (Zhu, 1983), warrant further exploration and
discussion.
The mechanics of seismic rotation may be related to various factors, including
nonlinear elasticity (Guyer and McCall, 1995; Guyer and Johnson, 1999), asymmetric
moment tensor (Teisseyre et al., 2003; Teisseyre, 2010), medium heterogeneity,
anisotropy (Pham et al., 2010; Sun et al., 2021), and site effects. This study focuses
only on isotropic and homogeneous media and three fundamental moment tensor
sources in the simulations of nonlinear small deformation. Therefore, the effect of
nonlinear geometric relation on wave propagation, especially for rotational
components, necessitates further investigation by testing the slipping angle, the shear
moment, the elastic parameters, and the anisotropy, among others. The current



discussion concentrates on wave propagation and the characteristics of 6-component
wavefields excited by three basic moment tensor sources to discuss the theoretical
approximation stemming solely from the linear assumption of small deformation, with
further analyses of other contributing factors slated for future research endeavors.
Observations and simulations of Taiwan Hualien earthquakes have verified the
existence of rotational motions along the northeast fault, resulting in prominent
Rayleigh-wave recordings and indicative of a vertical slipping mechanism in the
earthquake rupture process. In addition, the observation of stronger $\mathbf{R}_Z$ component and
two horizontal components suggests the presence of Love surface waves., signifying
clear horizontal slipping and torsion. This finding, aligning with Yu et al.'s (1999)
discovery, reveals the existence of horizontal rotational mechanisms within the
seismic belt of Taiwan attributed to the Pacific Plate beneath the Eurasian Plate from
the east, coupled with northward pressure exerted by the Philippines Sea Plate.
The simulations show the nonlinear effect cannot be neglected for near, regional,
and strong earthquakes, and that the rotational components observed at ground surface
will be stronger than the theoretical one, consistent with previous research.
Simulations in this study only portray the sources and medium in a simplified way.
The simulations of real earthquake scenarios present a much more intricate interplay
of source mechanisms and propagation mediums, encompassing long propagation
distances, and long time scales. So, the simulations of observed earthquakes,
especially for strong earthquakes, the nonlinear attributes through which seismic
waves couple with each other amplify the discrepancies arising from the nonlinear



assumption.

**6 Conclusions**
Based on seismic wave equations assuming linear small deformation, we have
derived elastic-wave equations that incorporate nonlinear part of Green strain tensor.
By numerical simulations in a three-dimensional full-space homogeneous medium
model using the finite difference method, our study discusses the distinctive
characteristics of translational and rotational motions elicited by three fundamental
moment tensor sources, shedding light on the wavefield differences between linear
and nonlinear assumptions. The following conclusions can be drawn from our study.
(1) Under the influence of the nonlinear Green tensor, the relative displacement,
deformation, and strain of spatial mass element in response to external forces are
superimposed with nonlinear second-order terms of strain tensor and rotation tensor,
resulting in third-order terms of displacement related to the shear and bulk moduli in
the propagation of elastic waves.
(2) Nonlinearity has a greater effect on ISO and CLVD sources than on DC
sources, and the effect of nonlinearity on the wavefield energy increases exponentially
with increasing magnitude. The nonlinear effect for ISO source primarily impacts S
waves. CLVD source generates wavefield difference ranging from 10 % to 20 % in
the 45° diagonal direction of P-wave front, similar to the anomalies caused by media
anisotropy.
(3) The errors caused by linearity approximation in rotations are more


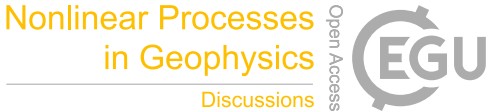

pronounced in pure basic seismic sources. Strong seismic events render the nonlinear
effect unbearable in simulations, underscoring the necessity of considering nonlinear
effects. In other cases, the linear approximation meets the accuracy requirements, so
the linear approximation can be used for relevant questions. Nonlinear small
deformation can be a factor in the rotational motion produced by strong earthquakes.
(4) The simulation of $\mathbf{E}1$ and $\mathbf{E}2$ primarily feature Rayleigh waves in vertical
translation and horizontal rotation. However, actual observations indicate a prevalent
existence of Love waves, potentially attributable to site effects or more complicated
focal mechanisms. The stronger-energy $\mathbf{E}2$ triggered relatively strong Love waves, so
its error caused by the resulting nonlinearity is larger.

**Author contributions.** WL: conceptualization, methodology, investigation, formal
analysis, writing - original draft. YW: conceptualization, writing - original draft and
revised draft. CC: in vestigation, formal analysis. LS: methodology.

**Data and resources.** The seismic records of $\mathbf{E}1$ are provided by the Institute of Earth
Sciences, Academia Sinica, Taiwan, China. The translational records of $\mathbf{E}2$ are
acquired from the Fujian Earthquake Agency.

**Competing interests.** The contact author has declared that neither of the authors has
any competing interes.

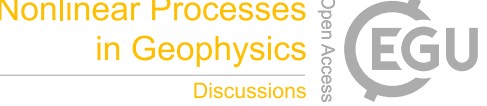

**Disclaimer.** Publisher's note: Copernicus Publications remains neutral with regard to jurisdictional claims made in the text, published maps, institutional affiliations, or any other geographical representation in this paper.

**Financial support.** This research is financially supported by the National Natural Science Foundation of China (No. 42150201、No. 62127815、No. U1839208).

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
