# Peer review of "under the nonlinearity in small deformation"

_Nonlinear Processes in Geophysics, 2024_

## Referee Comment (RC1)

https://doi.org/10.5194/npg-2024-17
Simulation characteristics of seismic translation 1and rotation under the assumption of nonlinear small deformation
by Wei Li, YunWang, Chang Chen, and Lixia Sun

REVIEW

The paper deals with a generally interesting topic, namely nonlinear effects on both the translational ground motion components
(which are the domain of traditional seismology), and on rotational components. Rotational seismology is currently a rapidly developing field,
hence the subject is highly topical. The authors investigate the numerical solution of the equation of motion with a strain tensor that includes some
nonlinear terms in addition to the linear ones and compare this with the 'classical' solution considering linear strain tensor that is commonly used
because of the small-deformation assumption. For numerical simulations they use a staggered grid finite difference scheme. The authors then
confront the simulations with the records of two selected Taiwan earthquakes.

Unfortunately the paper is not well written and will require significant refinement in a number of respects. The level of English is not good and often
makes it very difficult for the reader to understand the meaning. The quantities and equations should be better described in accordance to
mathematical conventions. The numerical models do not appear to be properly tested, and there is a lack of numerical error estimates.
Comparisons with real data from two selected Taiwan earthquakes are not convincing. The references show deficiencies or errors.
In my opinion, the paper requires MAJOR REVISION.

Comments

1. Introduction

Many reference deficiences - Grayzer, 1991, should be Graizer, 1991, moreover the corresponding reference in the list is not correct
Hua and Zhang (2002), in the list with the year 2022, moreover the citation cannot be traced with either of those years
Lee (2007) is missing in the reference list and it is not traceable anywhere on the internet
Chen et al. (2014) is not traceable on internet
(in the following I no longer list the deficiencies individually, but all references must be carefully checked and corrected
(many other deficiencies detected in the text !))

Line 35, the sentence "Seismic rotational motions are recorded in plenty of earthquakes..." - the sentence sounds somewhat exaggerated,
it gives the impression that registration of seismic rotational motions is quite common, which is not true. Rotational seismometry is constantly
evolving and there is still no universally applicable and reliable rotational sensor. Each rotation record must therefore be carefully analysed.

Lines 38-39, the term 'rotational torsions' is not defined.

Lines 47-48, "...observed rotations during strong ground motions exceed calculated translational components by one to two orders of magnitude"
Seismic rotations and translations represent totally different types of motion, measured in different units (e.g., m/s vs. rad/s). Thus, simple comparing their
amplitudes does not make sense. The above statement cannot even be supported by the literature.

Line 66 "... vertical rotation ..." What is it? Rotation about the vertical axis or rotation in a vertical plane? Please be specific.

2. Theories

(I don't derive the equations, so I can't comment on whether they are correct.)

Fig. 1 does not correspond to the explanatory text. Why the vectors dx and dx' are not shown in the figure (when mentioned in the text) and ds and ds'
are used instead? The notation is boldface indicating vectors, but they are scalars (defined as distances).

Line 91 "The following equations and tensors are written using the Kronecker symbol ..." I do not see the Kronecker symbol in any of the following
equations.

Eq. 1 ... i,j=x,y,z...  Strange notation, please write i=x,y,z and j=x,y,z (likewise in Eq. 13)

Line 98, notation inconsistency: uppercase X, Y, Z, but in Eq. 1 and below lowercase x,y,z (and likewise in the following text).

Line 131, using term 'small deformation' here (and likewise in the following text) is misleading in the given context as it implies neglecting the higher-order
terms,  including the second-order ones.

Eq. 10, mixing indices i,j,k and x,y,z, please clarify

Line 162, ... Rxz corresponds to..., but in Eq 11, there is no Rxz. Also boldface typing is inappropriate when denoting tensor components. Why the
uppercase symbol is used when we see lowercase in Eq. 4.? Please unify the notation.

Line 181, " ... to ensure the free-surface condition at the upper boundary ..." Under the free-surface condition, provided z is vertical, the rotational
components Rx and Ry simplify to individual space derivatives instead of their linear combinations (differences). This would be a good test of the
numerical approach adopted.

Line 182 (and Fig. 2) ... speaking about upper boundary implies that x is the vertical coordinate (not z, as usual)

Eq. 13 and the text below, there is no Mij in the equation. Please introduce Mij more strictly (from the mathematical point of view).

Line 191, a reader not familiar with seismic source represeantations may be surprised by 'force direction' and 'force arm'
without previously mentioning force pairs and dipoles.

3. Wavefield simulations ...

Figs. 3, 4 and 5: I'm not convinced that the non-linear effects in the figures are realistic, especially since this is a homogeneous isotropic model.
Rather, I suspect that we are seeing the effect of numerical errors. An analysis of the accuracy of the computational scheme used
is unfortunately lacking. However, the presence of numerical errors is evident, e.g., in the S waves for the ISO source. Since in the linear case
such a source does not generate any S waves, it is impossible for the relative difference to be finite. The analysis of numerical errors is extremely
important when dealing with amplitudes as small as those typical of seismic rotations. According to the theory, the rotation associated to P waves
must be zero in a homogeneous, isotropic, unbounded medium assuming a linear strain tensor. Again, the fact that we see a finite relative difference
between the linear and nonlinear cases is due to numerical errors. An analysis of the errors for the linear case could be made by comparison with
the analytical solution known for the homogeneous isotropic unbounded medium. Next, I think there is some problem with the time evolution of the
wavefield. How is it possible to see the S wave in the snapshots (when it is generated, e.g., by DC) if it propagates at 3 km/s and therefore
cannot reach the distance of 30 km in 8 seconds? Also, the wave between the P and S waveforms in Fig. 5 left is very suspicious. One last note: the
pictures show the amplitude values without specifying the 'source strength' (or initial amplitude).

Line 282, "...wavefield energy..." What exactly do you mean by this term? Give the corresponding formula.

Fig. 6b, For larger magnitudes (i.e., larger faults) the point source approximation may not be acceptable at the distance of 30 km.

4. Seismic observations and simulation ...

Line 313, ...(Wessel et al., 2019) refers only to the mapping tool. Please cite where the locations come from.

Line 313, ..."The receiver for E2...." Please give the name of the station (QS01?) and its instrumentation (the same for NA01 providing records of E1).

Line 315-316 (and Fig. 7b), "Additionally, a seismic array comprising seven 3C translational seismometers was deployed approximately 53 km from the epicenter
of E1". It is not clear wheather you use only the NA01 record or records from the whole array in your study. In that case provide at least array apperture and basic

instrument specifications. If the array is not used, Fig. 7b is irrelevant.

Eqs. 20, 21, Why new coordinates r,t,p are used? (not defined, not consistent with the previous equations and not used in further calculations)

Tab. 1 caption, "...observing stations". Plural form indicates that the same 1D structure is used for both stations (more than 300 km apart situated at
different 1-degree cells of the CRUST1.0 model). Is the structure so slowly changing to allow for that. Please comment on that.

Tab. 2, (53 km, :, 0 km) ? Shouldn't it be (53 km, 4 km, 0 km)?

Tab. 2, Grid interval 1 km is more than about 1/4 of the S-wavelength and about 1/6 of the P-wavelength. Isn't it too sparse grid for finite spatial differences
needed to evaluate rotations? (The situation is even much worse for model2 in Tab. 3)

Line 347-349, The general statement is true, simply speaking the higher the frequency is, the larger rotation amplitude we usually observe. However,
I do not understand the argumentation. The prevailing frequency in real records is higher (for body waves) than that of the synthetics, thus, no surprise
that we see larger rotation amplitude.

Fig. 8a - seems to be incorrect for three reasons: 1) strange dominat translational S-wave amplitude on z-component  (I would expect the largest S-wave
amplitude on y-componet, as it is almost in transverse direction), 2) Rx much larger that Ry. Acoording to Tab. 2 and the text bellow, x coordinate is
almost in the radial direction. Rotation about x thus should be negligible (at 53 km), Ry should be much larger! But Fig. 8a shows the opposite (no reason
for that in the simple 1D structure), 3) synthetic rotational components should have higher frequency content than the translational components (rotation
rate proportional to acceleration)

Fig 8b, (technical comment), It is not logical to show P- and S- onsets in iasp91 when the authors consider the CRUST1.0 model in their simulations.

Fig 8b, Horizontal (translational) components are more or less comparable (not significantly stronger) to the vertical one. Moreover the references
Abercombie (1997) and namely Guatteri et al., (2001) are totally inappropriate in the given context.

Fig 8b, (general comment), As mentioned above, none of the presently available rotational sensors can be considered as an 'ethalon' producing reliable
rotational records under all conditions. Therefore it is worth verifying the records whenever it is possible. In the case of NA01, there are even two
possibilities: ADR (array derived rotations) method utilizing Nanao array records, and  matching the rotational rate components to the related translational
acceleration components (for the frequency range considered it should be feasible). In case of a waveform mismatch, either the records are wrong or
the structure is significantly laterally inhomogeneous. In any case it would call into question any comparison of seismograms in Fig. 8a and 8b.

Tab. 3, Grid interval 5 km is comparable to the S wavelength and about one half of the P

wavelength, i.e., too big for spatial final differencing.

Tab. 3, (:, 0 km, 0 km) ? Shouldn't it be (100 km, 0 km, 0 km)? (Line 370)

Line 385-388, "That is consistent with previous studies that have argued that the observed rotational components have a relatively stronger amplitude
than the rotational component converted from translational components (Teisseyre et al., 2003)."
Teisseyre et al. (2003) make no such general claim.
They comment only on one case, where, moreover, the rotations were measured by a method which is nowadays outdated. What do you mean by
"converted from translational components"?

Fig. 9a - seems to be incorrect for several reasons: 1) high frequencies on Vz after 180 s (surface waves), the same for Ry, 2) synthetic rotational
components should be of a higher frequency content compared to the translational components (rotation rate proportional to acceleration),
3) prevailing frequency seems to be smaller than the declared 0.5 Hz (~ 0.3 Hz), 4) It's a pitty that the theoretical S-onset is not shown, I expect, it
is at about 90 s. It is again really strange that the S-wave amplitude is that much stronger on the vertical component than on the horizontal ones.

Fig. 9b, (technical comment), It is not logical to show P- and S- onsets in iasp91 when the authors consider the CRUST1.0 model in their simulations.

Fig. 9b, From where the rotational components come from? In case of the NA01 station, blueSeis rotational sensor was mentioned, but no mention about
any rotational sensor in the station that records E2.

Fig. 9b, As expected, rotational components are more high frequency than translational ones. Nevertheless, it would be worth 'verifying' rotational
records by matching waveforms to the relevant acceleration components.

Line 408 and Fig. 10, "...Vx and Vy components, with errors up to 10 %...." Fig. 10b shows even much bigger errors: Vx reaching 15% and Vy probably
significantly exceeding 15%! I consider this to be unrealistic taking into account the distance (327 km, far from the earthquake focal zone) and simplicity
of the structute (isotropic 1D model with homogeneous layers). If we believed that is true, we would have to question all existing seismology!

Line 417-419, I do not understand the argumentation - speaking about Fig. 10 we speak about relative errors (in %) in the synthetic example,
it has nothing to do with an amplitude decay on rotational components.

5. Discussions and 6. Conclusions

I do not consider the claims made here to be relevant until the analyses, tests and corrections suggested above have been carried out.

General comment: the influence of much more significant factors such as lateral structure inhomogeneity, anisotropy, attenuation, etc. should be

investigated before any consideration of nonlinear effects in real seismograms based on simple simulations in 1D models composed of homogeneous layers
filled with isotropic and perfectly elastic material.

---

## Referee Comment (RC2)

**Review of the paper "Simulation characteristics of seismic translation and rotation under the assumption of nonlinear small deformation" by**

Wei Li[1,2], Yun Wang[1,2], Chang Chen[1,2] and Lixia Sun[1,3]

[1] *"MWMC" group, School of Geophysics and Information Technology, China University of Geosciences, Beijing 100083, China*

[2] *State Key Laboratory of Geological Processes and Mineral Resources, China University of Geosciences, Beijing 100083, China*

[3] *Sinopec Research Institute of Petroleum Engineering Co., Ltd., Beijing 102206, China*

This is the second time that I review this paper, and just like before I have problems with the theoretical basis of this work. I do recognize that the authors present an interesting study of the *possible* non-linear elastic effects observed in seismological data, but the paper needs solid and clear theoretical basis as well as the main message of the contribution. In the next I elaborate the theoretical basis.

**Theoretical basis**    The authors begin with a good explanation of the strain tensor and its meaning as a distance measure. They mention the non-linear term that is often avoided in research papers, which is usually done in the assumption that one is far away from the seismic source.

The expression for the strain tensor including high-order terms is given by

$$E_{ij} = \frac{1}{2}\left(\partial_i u_j + \partial_j u_i + \partial_i u_j \partial_j u_i\right) \quad \text{with} \quad i,j \in \{1,2,3\}. \tag{1}$$

This is equation (2) in the paper. Until this point everything is fine, however soon enough the authors star with a zoo of equations and terminology that is irrelevant for the study. For example, the authors re-write eq. (2) as eq. (5) (what for?) and then introduce eqs. (6)–(7) and (8) and start to talk about volumetric strains (what for?) to re-write the equation of motion in eq. (9). What is the need to talk about volumetric strains? write something like eq. (8)? what for? In addition to that, the authors write eq. (10) combining two notations $(x,y,z)$ and $(1,2,3)$. This is a complete nonsense. Again, the authors write the equation of motion in eq. (9) using eq. (8), combining two notation, such a huge large equation as given in eq. (8). What is the need of such a huge bad notation and complication?

Please note that the first line of eq. (10) uses $i,j$ so one assumes that $i,j \in \{1,2,3\}$ and in the next line of the same equation one reads $i,j,k$ and $x,y,z$. One can again assume that $i,j,k \in \{1,2,3\}$ and some direct analogy to $x,y,z$. But that all. This is a mathematical error in notation and the huge confusion that this brings into the study just kills every possible contribution.

Having pointed out that there ar mathematical errors/confusions in the equations, I do not want to even look to the numerical implementation. In addition, I have said this, the numerical implementation has to be accompanied by a benchmark against another well tested numerical code. There many that can be used, for instance SPECFEM (Komatitsch and Vilotte, 1998), the codes from the group of Prof. Peter Moczo, etc. I do understand that these codes are discretizing the linear approximation. The idea should be that the code that the authors present should be benchmarked in the linear approximation with these well known codes (one of them) and later use the code to draw differences and similarities with the non-linear case.

I do not look forward to give a bad review one more time for this paper. For this reason I will write here the mathematical basis that this paper needs. One will see that eqs. (6)–(7) and (8) are completely unnecessary and confusing. Eqs. (11) should not be written that way... not even mentioning eq. (12) shouldn't be either.

**Theoretical basis (clarifications)**    If one looks to find the equations of motion related to any linear and symmetric strain measure, we can simply write the equations of motion as follows (for a good introduction to the topic see Slawinski (2010))

$$\rho \underbrace{\partial_t^2 u_j}_{\text{acceleration term}} = \underbrace{\partial_i \sigma_{ij}}_{\text{Divergence of the stress}} \quad \text{with} \quad i,j \in \{1,2,3\}, \tag{2}$$

where $\rho$ is the material density, $u$ the displacement vector and $\sigma_{ij}$ is the (second-order) stress tensor defined as

$$\sigma_{ij} = \mathbb{C}_{ijkl} E_{kl}, \tag{3}$$

where $\mathbb{C}$ is a fourth-order tensor of elastic constants with the symmetries $\mathbb{C}_{ijkl} = \mathbb{C}_{jikl} = \mathbb{C}_{ijlk} = \mathbb{C}_{klij}$ (Slawinski, 2010) and $E_{kl}$ the chosen strain measure. Note that we use the conventional notation $\{1, 2, 3\}$ for $\{x, y, z\}$.

If one assumes isotopic symmetry for the elastic tensor $\mathbb{C}_{ijkl}$, we can write (Dahlen and Tromp, 1998)

$$\mathbb{C}_{ijkl} = \lambda \delta_{ij} \delta_{kl} + \mu(\delta_{ik} \delta_{jl} + \delta_{il} \delta_{jk}) \quad \text{with} \quad i, j, k \in \{1, 2, 3\}, \tag{4}$$

where $\lambda, \mu$ are Lamé parameters and $\delta$ the Dirac distribution. If we use the conventional linear strain tensor as given by the following expression

$$E_{ij} = \frac{1}{2} \left( \partial_i u_j + \partial_j u_i \right) \quad \text{with} \quad i, j \in \{1, 2, 3\}, \tag{5}$$

we arrive to the well known equation (doing step by step!)

$$
\begin{aligned}
\partial_t^2 u_j &= \partial_i \left( \mathbb{C}_{ijkl} E_{kl} \right) \\
\partial_t^2 u_j &= \partial_i \left( \lambda \delta_{ij} \delta_{kl} \frac{1}{2} \left( \partial_k u_l + \partial_l u_k \right) + \mu(\delta_{ik} \delta_{jl} + \delta_{il} \delta_{jk}) \frac{1}{2} \left( \partial_k u_l + \partial_l u_k \right) \right) \\
\partial_t^2 u_j &= \partial_i \left( \lambda \delta_{ij} \partial_k u_k + \mu(\delta_{ik} \delta_{jl} + \delta_{il} \delta_{jk}) \frac{1}{2} \partial_k u_l + \mu(\delta_{ik} \delta_{jl} + \delta_{il} \delta_{jk}) \frac{1}{2} \partial_l u_k \right) \\
\partial_t^2 u_j &= \partial_i \left( \lambda \delta_{ij} \partial_k u_k + \mu \frac{1}{2} \partial_i u_j + \mu \frac{1}{2} \partial_j u_i + \mu \frac{1}{2} \partial_j u_i + \mu \frac{1}{2} \partial_i u_j \right) \\
\partial_t^2 u_j &= \partial_i \left( \lambda \delta_{ij} \partial_k u_k + \mu \left( \partial_i u_j + \partial_j u_i \right) \right) \quad \text{with} \quad i, j \in \{1, 2, 3\}.
\end{aligned} \tag{6}
$$

If we consider the strain tensor defined as in eq. (1), just simply need to add the contribution of $\partial_i u_j \partial_j u_i$ to the previous equation, i.e.,

$$
\begin{aligned}
\left[ \lambda \delta_{ij} \delta_{kl} + \mu(\delta_{ik} \delta_{jl} + \delta_{il} \delta_{jk}) \right] \partial_k u_l \partial_l u_k &= \lambda \delta_{ij} \delta_{kl} \partial_k u_l \partial_l u_k + \mu(\delta_{ik} \delta_{jl} + \delta_{il} \delta_{jk}) \partial_k u_l \partial_l u_k \\
&= \lambda \delta_{ij} \partial_k u_k \partial_k u_k + 2\mu \partial_i u_j \partial_j u_i
\end{aligned} \tag{7}
$$

Thus, the equation of motion related to strain measure given in eq.(1) becomes

$$\partial_t^2 u_j = \partial_i \left( \lambda \delta_{ij} \left( \partial_k u_k + \partial_k u_k \partial_k u_k \right) + \mu \left( \partial_i u_j + \partial_j u_i + 2 \partial_i u_j \partial_j u_i \right) \right) \quad \text{with} \quad i, j, k \in \{1, 2, 3\}. \tag{8}$$

No that the equation of motion is symmetric, we can simply interchange $j \to j$ to obtain a more familiar expression as follows

$$\partial_t^2 u_i = \partial_j \left( \lambda \delta_{ij} \left( \partial_k u_k + \partial_k u_k \partial_k u_k \right) + \mu \left( \partial_i u_j + \partial_j u_i + 2 \partial_i u_j \partial_j u_i \right) \right) \quad \text{with} \quad i, j, k \in \{1, 2, 3\}. \tag{9}$$

Equation (9) uses a single notation for the coordinates $\{1, 2, 3\}$ and Einstein notation for repeated summation. For Finite-Difference calculation it is useful to write the equation of motion (9) in its displacement (or velocity)–stress notation as follows

$$\partial_t^2 u_i = \partial_j \sigma_{ji}, \tag{10}$$

with the stress tensor defined as

$$\sigma_{ji} = \left( \lambda \delta_{ij} \left( \partial_k u_k + \underbrace{\partial_k u_k \partial_k u_k}_{\text{additional term}} \right) + \mu \left( \partial_i u_j + \partial_j u_i + \underbrace{2 \partial_i u_j \partial_j u_i}_{\text{additional term}} \right) \right). \tag{11}$$

Eq. (9) is the equation that should be analyzed in the paper and its velocity-stress formulation in the numerical implementation. The additional terms should be properly understood.

**Conclusion** I therefore cannot recommend this paper for publication as it is. It has confusing mathematical basis which do not allow to evaluate the correctness of the theory and the numerical predictions and in addition, the English is not well written. I understand that writing scientific English is not an easy task, but there are services offered on the web that one can use to write a sufficiently good English for a publication. One can read expressions like: *nonlinear effect unbearable in simulations...* (see line 496).

The authors show a potential interesting application but the paper needs substantial work to be done and once again it needs clear and solid mathematical basis with at least one benchmark of the code that are using.

The reviewer

**References**

Dahlen, F. and Tromp, J. (1998). *Theoretical global seismology*. Princeton university press.

Komatitsch, D. and Vilotte, J.-P. (1998). The spectral element method: an efficient tool to simulate the seismic response of 2D and 3D geological structures. *Bulletin of the Seismological Society of America*, 88(2):368–392.

Slawinski, M. (2010). *Waves and Rays in Elastic Continua*. World Scientific Publishing Company.

---

## Author Response (AR1)

**Simulation characteristics of seismic translation and rotation under the nonlinearity in small deformation**

Wei Li [1,2,], Yun Wang [1,2,*], Chang Chen[1,2], Lixia Sun[1,3]

[1] *"MWMC" group, School of Geophysics and Information Technology, China University of Geosciences, Beijing 100083, China*

[2] *State Key Laboratory of Geological Processes and Mineral Resources, China University of Geosciences, Beijing 100083, China*

[3] *Sinopec Research Institute of Petroleum Engineering Co., Ltd., Beijing 102206, China*

[*] Corresponding author: wangyun@mail.gyig.ac.cn.

**Replies to Reviewer#1**

**1. Introduction**

**1:** Many reference deficiences - Grayzer, 1991, should be Graizer, 1991, moreover the corresponding reference in the list is not correct Hua and Zhang (2002), in the list with the year 2022, moreover the citation cannot be traced with either of those years Lee (2007) is missing in the reference list and it is not traceable anywhere on the internet Chen et al. (2014) is not traceable on internet (in the following I no longer list the deficiencies individually, but all references must be carefully checked and corrected    (many other deficiencies detected in the text !))

Thank you for the comments. We have corrected the errors and omissions, carefully checked all references, and corrected any other potential deficiencies found.

**2:** Line 35, the sentence "Seismic rotational motions are recorded in plenty of earthquakes..." - the sentence sounds somewhat exaggerated, it gives the impression that registration of seismic rotational motions is quite common, which is not true.

Rotational seismometry is constantly evolving and there is still no universally applicable and reliable rotational sensor. Each rotation record must therefore be carefully analysed.

Thank you for the comments. We have revised it to a more cautious wording to better reflect the current situation more accurately.

**3:** Lines 38-39, the term 'rotational torsions' is not defined.

Thank you for the comment. The term "rotational torsions" refers to the rotation around the vertical axis (rotational z-component) caused by seismic waves. It shouldn't refer to just torsion, so we changed it to "rotation" here., which we have explained in the manuscript.

**4:** Lines 47-48, "...observed rotations during strong ground motions exceed calculated translational components by one to two orders of magnitude" Seismic rotations and translations represent totally different types of motion, measured in different units (e.g., m/s vs. rad/s). Thus, simple comparing their amplitudes does not make sense. The above statement cannot even be supported by the literature.

Thank you for the comments. We are not referring to a comparison of the magnitude of the translational and rotational components, but to a comparison between directly observed rotational motion and rotational motion indirectly derived from arrayed translational components, especially in the case of strong and near earthquake, where the directly observed rotational motion is 1 - 2 orders of magnitude stronger than the indirectly observed rotational motion. We have improved our wording to avoid any potential misinterpretation.

**5:** Line 66 "... vertical rotation ..." What is it? Rotation about the vertical axis or rotation in a vertical plane? Please be specific.

Thank you for the comments. We made it clear in the text that 'vertical rotation' refers to rotation about the vertical axis.

**2. Theories**

**6:** Fig. 1 does not correspond to the explanatory text. Why the vectors dx and dx' are not shown in the figure (when mentioned in the text) and ds and ds' are used instead? The notation is boldface indicating vectors, but they are scalars (defined as distances).

Thank you for the comments. We have updated Figure 1 to be consistent with the description in the text and corrected inappropriate notation.

**7:** Line 91 "The following equations and tensors are written using the Kronecker symbol ..." I do not see the Kronecker symbol in any of the following equations.Eq. 1 ... i,j=x,y,z... Strange notation, please write i=x,y,z and j=x,y,z (likewise in Eq. 13) Line 98, notation inconsistency: uppercase X, Y, Z, but in Eq. 1 and below lowercase x,y,z (and likewise in the following text).

Thank you for the comments. We have revised the use of symbols in the equations and have used uniform characters to represent the coordinate axes.

**8:** Line 131, using term 'small deformation' here (and likewise in the following text) is misleading in the given context as it implies neglecting the higher-order terms, including the second-order ones.

Thank you for the comments. ~~We understand the possible misunderstanding of the term "small deformation". Indeed, in conventional understanding, it is often associated with the hypothesis of small linear deformation that ignores higher-order terms. To avoid misunderstanding, we will seek to make this point more explicit in our manuscript and use more precise wording to ensure that readers can understand the content of our study and the theoretical framework used.~~

**9:** Eq. 10, mixing indices i,j,k and x,y,z, please clarify

Thank you for the comment. We have rewritten and corrected the use of symbols to use consistent characters to represent the axes.

**10:** Line 162, ... Rxz corresponds to..., but in Eq 11, there is no Rxz. Also boldface typing is inappropriate when denoting tensor components. Why the uppercase symbol is used when we see lowercase in Eq. 4.? Please unify the notation.

Thank you for the comments. We have modified the formulas, removed the inappropriate bold and standardized the capitalization of physical quantities.

**11:** Line 181, " ... to ensure the free-surface condition at the upper boundary ..." Under the free-surface condition, provided z is vertical, the rotational components Rx and Ry

simplify to individual space derivatives instead of their linear combinations (differences). This would be a good test of the numerical approach adopted.

Thank you for the comments. The acoustic boundary replacement method is widely accepted for efficiently dealing with the reflection of seismic waves at free surfaces and accurately modeling the effect of the ground surface. We didn't use the free-surface rotational motion definition equation because we obtained the numerical solutions for both translational and rotational motion simultaneously at all model grid points using the acoustic boundary method. Based on the setting formula of the free surface conditions, the influence is applied to the simulation, and numerical solutions at the free surface are obtained. Of course, this is indeed a good test method, and we will test both free boundary simulation methods further to verify the validity and reliability of the results.

**12:** Line 182 (and Fig. 2) ... speaking about upper boundary implies that x is the vertical coordinate (not z, as usual)

Thank you for the comment. The upper boundary is a common expression for this model, which we will describe in the text as setting the free surface conditions at the corresponding position of the z-axis of the model according to setting formulas.

**13:** Eq. 13 and the text below, there is no Mij in the equation. Please introduce Mij more strictly (from the mathematical point of view).

Thank you for the comments. We have optimized the introduction of Mij.

**14:** Line 191, a reader not familiar with seismic source represeantations may be surprised by 'force direction' and 'force arm' without previously mentioning force pairs and dipoles.

Thank you for the comments. We agree and have removed this potentially unnecessary introduction.

**3. Wavefield simulation**

**15:** Figs. 3, 4 and 5: I'm not convinced that the non-linear effects in the figures are realistic, especially since this is a homogeneous isotropic model. Rather, I suspect that we are seeing the effect of numerical errors. An analysis of the accuracy of the computational scheme used is unfortunately lacking. However, the presence of

numerical errors is evident, e.g., in the S waves for the ISO source. Since in the linear case such a source does not generate any S waves, it is impossible for the relative difference to be finite. The analysis of numerical errors is extremely important when dealing with amplitudes as small as those typical of seismic rotations. According to the theory, the rotation associated to P waves must be zero in a homogeneous, isotropic, unbounded medium assuming a linear strain tensor. Again, the fact that we see a finite relative difference between the linear and nonlinear cases is due to numerical errors. An analysis of the errors for the linear case could be made by comparison with the analytical solution known for the homogeneous isotropic unbounded medium. Next, I think there is some problem with the time evolution of the wavefield. How is it possible to see the S wave in the snapshots (when it is generated, e.g., by DC) if it propagates at 3 km/s and therefore cannot reach the distance of 30 km in 8 seconds? Also, the wave between the P and S waveforms in Fig. 5 left is very suspicious. One last note: the pictures show the amplitude values without specifying the 'source strength' (or initial amplitude).

Thank you for the comments. We would like to clarify and respond to the concerns regarding Figures 3, 4, and 5

**(1) Comparative validation and error analysis:** After the comparison of numerical errors, the numerical errors of the current simulation parameters are very small and the results are more accurate. About benchmark, time is limited we haven't done a good benchmark comparison, but we will continue to work on it here.

**(2) S-wave propagation time**: We have re-simulated and resized the model and show the snapshots in a different way,pleas see Figures 3, 4, and 5.

the figure has not yet reached the observation point 30 km away. The misrepresentation here may be due to poorly selected or labeled time slices in the graphical display. We will readjust the time slices to ensure a clear presentation.

**(3) S-waves in ISO source simulations:** We agree that ISO sources do not generate S-waves, and according to linear theory, there is only P-wave propagation in homogeneous isotropic media due to the volume change associated with a pure pressure field. In numerical simulations, due to the inherent presence of numerical errors, the numerical results may show non-zero values even at locations where no S-wave propagation is theoretically expected. This situation can lead to bias in the comparison results when the relative change rate is subsequently calculated. Therefore, we will actively seek a more accurate and reliable comparison method to ensure the accuracy and reliability of the results. his method of comparison is not appropriate and we have removed it from the manuscript.

**(4) P-waves in rotational component**: we infer that: perhaps the loading of the force source and the incomplete symmetry of the staggered mesh produce very weak P-waves. In addition, the situation becomes complicated when the medium has nonlinear properties, and the nonlinear effects perhaps produce small vibrations similar to P-waves, faintly visible on the rotational component of the elastic medium simulation, but actually very weak. Indeed, P-waves are usually described as pure compressional waves. Numerical approximations and discretization grids during simulation may introduce small unphysical effects. These effects may become significant at high accuracy requirements, or perhaps the forces are loaded in such a way that there are residual wavefields that lead to the observed P waves of the rotational component signals. We will make further investigation on this.

**(5) Source intensity description**: We used a source of magnitude 6 in the pre-revision manuscript and a source of magnitude 7 in the revised simulation. This is our oversight, and we will clearly introduce the source

**16:** Line 282, "...wavefield energy..." What exactly do you mean by this term? Give the corresponding formula.

Thank you for the comment. We have added the wavefield energy calculation formula. Regarding the 'wavefield energy', it is an approximate total energy by calculating the square of the wavefield value at each grid point and summing, which is calculated by the following formula:

$$E = \sum_{i,j,k} \left| u_{i,j,k} \right|^2 \Delta V_{i,j,k}$$

where $u_{i,j,k}$ is the wavefield value (which can be a physical quantity such as displacement or velocity) at the grid point (i,j,k) and $\Delta V_{i,j,k}$ is the volume of the unit grid. This calculation is based on the fundamental principle that the energy is proportional to the square of the amplitude, and we use it as an approximation for the wavefield energy in discretized grid systems.

**17:** Fig. 6b, For larger magnitudes (i.e., larger faults) the point source approximation may not be acceptable at the distance of 30 km.

Thank you for the comments. We understand that in practice the sources of large earthquakes usually have complex geometries and slip distributions, and these properties cannot be adequately characterized in point source approximation. However, our work focuses on the impact of nonlinear effects on seismic waves rather than on accurately describing the sources themselves. Therefore, we chose the point-source approximation as a starting point that is reasonable and effective given the goals of this study. We are also aware of the limitations of the point-source approximation, which we will explain in the discussion section. In future research, we plan to further explore more complex seismic source models (such as finite fault models) to more accurately evaluate the performance of nonlinear effects under different seismic source conditions.

**18:** Line 313, ...(Wessel et al., 2019) refers only to the mapping tool. Please cite where the locations come from.

Thank you for the comment. We have made it clear in the text.

**19:** Line 313, ..."The receiver for E2...." Please give the name of the station (QS01?) and its instrumentation (the same for NA01 providing records of E1).

Thank you for the comment. We use Blueseis in the QS01 station for E2, which we specified in the revised manuscript.

**20:** Line 315-316 (and Fig. 7b), "Additionally, a seismic array comprising seven 3C translational seismometers was deployed approximately 53 km from the epicenter of E1". It is not clear whether you use only the NA01 record or records from the whole array in your study. In that case provide at least array apperture and basic instrument specifications. If the array is not used, Fig. 7b is irrelevant.

Thank you for the comments. We analyzed the seismic data from the NA01 station, not the entire seismic array. Figure 7b shows the array deployment, which provides the experimental context, not the direct object of analysis. We have clarified this information in the relevant part of the text and made appropriate changes to Figure 7b to ensure that readers are not misled.

**21:** Eqs. 20, 21, Why new coordinates r,t,p are used? (not defined, not consistent with the previous equations and not used in further calculations)

Thank you for the comment. We have represented these tensor components as expressions of the coordinates x, y, z.

**22:** Tab. 1 caption, "...observing stations". Plural form indicates that the same 1D structure is used for both stations (more than 300 km apart situated at different 1-degree cells of the CRUST1.0 model). Is the structure so slowly changing to allow for that. Please comment on that.

Thank you for the comments. Regarding the use of the same one-dimensional (1D) structure model for two stations more than 200 km apart in this study, we have the following main considerations and we have given a brief account of this in the revised manuscript :

**(1) Aim of the study and simplification:** Our main objective is to study the

effect of different source mechanisms and seismic wave propagation distances on the nonlinear errors of seismic records by simulating the two earthquakes. To reduce the influence of other factors, we decided to use the same stratigraphic model in the simulation. Although there may be differences in the details of the actual media, we are concerned with the nonlinear effects during seismic wave propagation rather than the local details of the stratigraphic structures. These differences are not a major comparative factor for the nonlinear features of interest in our work, and this simplification has been adopted given the relative stability of the structures at larger scales.

**(2) Stability of crustal structure:** Crustal structure changes relatively slowly on macroscopic scales (e.g., a few hundred kilometers), especially in regions without significant influence of geological activity. Therefore, it is reasonable to assume that the two observatories are in a similar crustal structure context, despite their distance from each other.

We are also aware of possible differences in crustal structure at the microscopic scale and the effect that these differences may have on the propagation characteristics of seismic waves. In future studies, we plan to further refine the crustal model to account for differences in crustal structure between different observatories and to more fully assess their impact on the nonlinear effects in seismic records.

**23:** Tab. 2, (53 km, :, 0 km) ? Shouldn't it be (53 km, 4 km, 0 km)?

Thank you for the comment. Our simulation convention is generally to stores seismic records along a line of measurement in a certain direction, so we wrote this before, now we have changed this to (53 km, 4 km, 0 km) to show the receiver location.

**24:** Tab. 2, Grid interval 1 km is more than about 1/4 of the S-wavelength and about 1/6 of the P-wavelength. Isn't it too sparse grid for finite spatial differences needed to evaluate rotations? (The situation is even much worse for model2 in Tab. 3)

Thank you for the comments. Currently, we use a 1-km grid interval for E1 and 2-km grid interval E2 simulations, and their grid accuracy is acceptable in terms of

numerical errors over the deeper layers, and the shallowest layer impacts the overall study objective less. Our current staggered-grid finite-difference method still uses uniform grid intervals, and we are considering improving it to a variable grid to accommodate thinner layer.

**25:** Line 347-349, The general statement is true, simply speaking the higher the frequency is, the larger rotation amplitude we usually observe. However,I do not understand the argumentation. The prevailing frequency in real records is higher (for body waves) than that of the synthetics, thus, no surprise that we see larger rotation amplitude.

Thank you for the comments. Observed data tend to be richer in high-frequencysignals than synthetic data with a single dominant frequency, and this can indeed be an important factor in the difference in magnitude between the observed translational and rotational magnitudes and the synthetic translational and rotational amplitudes. Although the propagation medium is elastic in the simulations, the real medium tends to be viscoelastic, leading to absorption of seismic energy and attenuation of high-frequency signals. Furthermore, the mechanism behind the phenomenon of more pronounced rotational signals at high frequencies may involve many complex factors, such as the source characteristics, the propagation path, and the local geological conditions at the receiver site. The interplay of these factors provides new perspectives and rich scientific research space for the in-depth study of high-frequency rotational signals, which is worthy of further exploration in future work. We have modified these analysis.

**26:** Fig. 8a - s eems to be incorrect for three reasons: 1) strange dominant translational S-wave amplitude on z-component (I would expect the largest S-wave amplitude on y-component, as it is almost in transverse direction), 2) Rx much larger that Ry. According to Tab. 2 and the text bellow, x coordinate is almost in the radial direction. Rotation about x thus should be negligible (at 53 km), Ry should be much larger! But Fig. 8a shows the opposite (no reason for that in the simple 1D structure), 3) synthetic rotational components should have higher frequency content than the translational

components (rotation rate proportional to acceleration)

Thank you for the comments. When we simulated with the new formula, after checking, we found that the E1 model with overall north direction was set to south direction, which has been corrected (please see Figure 9). The Vz still has a large amplitude, probably due to the interaction of the seismic wave with the inhomogeneous layer structure, and the amplitude in the Ry component is the largest in the rotational one. As for the frequency of the rotational component, the frequency of the rotational component should indeed be higher than that of the translational motion, but when the source is loaded with a single and lower frequency signal, the higher frequency of the rotation will not be very prominent.

**27:** Fig 8b, (technical comment), It is not logical to show P- and S- onsets in iasp91 when the authors consider the CRUST1.0 model in their simulations.

Thank you for the comment. In the revised manuscript, we have removed the P- and S- onsets in iasp91.

**28:** Fig 8b, Horizontal (translational) components are more or less comparable (not significantly stronger) to the vertical one. Moreover the references Abercombie (1997) and namely Guatteri et al., (2001) are totally inappropriate in the given context.

Thank you for the comments. We have modified our statements and re-evaluated the references to Abercombie (1997) and Guatteri et al. (2001) to ensure their applicability.

**29:** Fig 8b, (general comment), As mentioned above, none of the presently available rotational sensors can be considered as an 'ethalon' producing reliable rotational records under all conditions. Therefore it is worth verifying the records whenever it is possible. In the case of NA01, there are even two possibilities: ADR (array derived rotations) method utilizing Nanao array records, and matching the rotational rate components to the related translational acceleration components (for the frequency

range considered it should be feasible). In case of a waveform mismatch, either the records are wrong or the structure is significantly laterally inhomogeneous. In any case it would call into question any comparison of seismograms in Fig. 8a and 8b.

Thank you for the comments. We fully understand your concern about the reliability of the data recorded by the rotation sensors and agree with the importance of validating the records. For the E1 observation record, we have compared the ADR-derived rotational motions with the rotational motions directly observed at the NA01 station, and the results are in good agreement (please see the following figure). Although both methods have their strengths and limitations, deriving rotational motions from translational seismometers involves approximations and may introduce errors and the direct measurement is a more direct and potentially less error-prone way to capture rotational motions, especially in cases where the structural integrity and homogeneity of the observation site is well established.

[Figure]

**30:** Tab. 3, Grid interval 5 km is comparable to the S wavelength and about one half of the P wavelength, i.e., too big for spatial final differencing.

Thank you for the comments. We have modified the grid interval for the E2 simulation to 2km and resimulated E2.

**31:** Tab. 3, (:, 0 km, 0 km) ? Shouldn't it be (100 km, 0 km, 0 km)? (Line 370)

Thank you for the comment. We have changed it to the receiver location of (100 km,

0 km, 0 km).We have changed this to make it clear that an observation system is along the entire line in this dimension.

**32:** Line 385-388, "That is consistent with previous studies that have argued that the observed rotational components have a relatively stronger amplitude than the rotational component converted from translational components (Teisseyre et al., 2003)." Teisseyre et al. (2003) make no such general claim. They comment only on one case, where, moreover, the rotations were measured by a method which is nowadays outdated. What do you mean by "converted from translational components"?

Thank you for the comments. We have corrected the citation. Lee (2007) pointed out that the directly recorded rotational motions in Japan and Taiwan under strong and near-field earthquakes are one to two orders of magnitude larger than the rotational motions derived indirectly from translational accelerometer arrays. This difference may be partly due to errors introduced during the differential calculation process, butitis more likely mainly due to complex factors such as nonlinear elastic or site conditions at the receiving point affecting the seismic rotation. The inaccurate expression in the original manuscript, "converted from translational components," refers to the rotational motion derived from records of translational accelerometer arrays, which we have corrected. We have corrected the citation and modified the application and inappropriate expressions.

**33:** Fig. 9a - seems to be incorrect for several reasons: 1) high frequencies on Vz after 180 s (surface waves), the same for Ry, 2) synthetic rotational components should be of a higher frequency content compared to the translational components (rotation rate proportional to acceleration), 3) prevailing frequency seems to be smaller than the declared 0.5 Hz (~ 0.3 Hz), 4) It's a pitty that the theoretical S-onset is not shown, I expect, it is at about 90 s. It is again really strange that the S-wave amplitude is that much stronger on the vertical component than on the horizontal ones.

Thank you for the comments. We will adjust the parameters, and after adjustment, we will rerun the simulation and analyze the results to obtain more reliable and convincing results. We will seek reasonable explanations and justifications if the

adjusted simulation results still show "unintended" or "strange" phenomena. We re-simulated the results after the grid spacing adjustment; please see Figure 11.

**34:** Fig. 9b, (technical comment), It is not logical to show P- and S- onsets in iasp91 when the authors consider the CRUST1.0 model in their simulations. Fig. 9b, From where the rotational components come from? In case of the NA01 station, blueSeis rotational sensor was mentioned, but no mention about any rotational sensor in the station that records E2.

Thank you for the comments. We have removed the P-waves and S-waves onsets in revised version. For recording E2, we use blueSeis to record rotational motions.

**35:** Fig. 9b, As expected, rotational components are more high frequency than translational ones. Nevertheless, it would be worth 'verifying' rotational records by matching waveforms to the relevant acceleration components.

Thank you for the comments. The E2 observation station QS01 is located inside a cave at the Qingyuan Mountain Seismic Station in Quanzhou, Fujian. This location provides a relatively stable and low-noise environment, which is beneficial for the accuracy and reliability of the rotational recordings. In subsequent work, we will consider matching the rotational waveforms with the relevant acceleration components for further verification.

**36:** Line 408 and Fig. 10, "...Vx and Vy components, with errors up to 10 %...." Fig. 10b shows even much bigger errors: Vx reaching 15% and Vy probably significantly exceeding 15%! I consider this to be unrealistic taking into account the distance (327 km, far from the earthquake focal zone) and simplicity of the structure (isotropic 1D model with homogeneous layers). If we believed that is true, we would have to question all existing seismology!

Thank you for the comments. Considering the distance of the observation site from the epicenter (327 km) and the fact that we used a relatively simple model (1D isotropic homogeneous layered model), this error level was indeed beyond our initial expectations and prompted us to re-examine the situation. We plan to validate and compare the simulation results by analyzing the error and improving the model. If the error remains high, we will investigate possible physical mechanisms or unidentified

confounding factors and conduct more in-depth research and validation before reaching a reliable conclusion to ensure our work adds value. The re-simulation results of the E2 show that there is still a relatively significant error in the six components, which can only mean that in an ideal isotropic elastic medium, with a strong magnitude, the nonlinear error produces a really large error. But for most earthquakes of microseismic, small and moderate earthquakes and far-field observations, the theoretical nonlinear errors are minimal, and in actual observations, the influence of the nonlinear terms should be far less than the theoretical simulations.

**37:** Line 417-419, I do not understand the argumentation - speaking about Fig. 10 we speak about relative errors (in %) in the synthetic example, it has nothing to do with an amplitude decay on rotational components.

Thank you for the comments. We want to emphasize that, whether in simulations or actual observations, the rotational component in far-field records of natural earthquakes typically shows a faster amplitude decay trend than the translational component. For this reason, the accumulation of nonlinear errors in the rotational component may be relatively small. When discussing the relative errors in Figure 10, we did not directly address the issue of amplitude decay of the rotational component. We will modify relevant expressions in the manuscript to ensure that they accurately convey our research content and findings. We have removed the less appropriate analyses here.

**5. Discussions and 6. Conclusions**

**38:** I do not consider the claims made here to be relevant until the analyses, tests and corrections suggested above have been carried out. General comment: the influence of much more significant factors such as lateral structure inhomogeneity, anisotropy, attenuation, etc. should be investigated before any consideration of nonlinear effects in real seismograms based on simple simulations in 1D models composed of homogeneous layers filled with isotropic and perfectly elastic material.

Thank you for your thorough review and valuable comments on our research manuscript. We fully understand that studying the nonlinear effects of other key factors in depth is necessary. These tasks are indeed necessary and important for our

understanding of nonlinear wave propagation. After consideration, we will rewrite the conclusion and discussion of the manuscript to point out the directions for further research that we need to do in the future, especially your suggestion to analyze and consider more significant factors. We are well aware that there are still shortcomings in the manuscript, but your valuable opinions have pointed us in the direction of improvement. We sincerely thank the reviewers for their attention and contribution to our work, and we will revise according to the above suggestions.

We look forward to presenting more accurate and complete research results in subsequent version. For language issues, we will seek professional language services to improve the English expression to ensure the readability and academic standards of the manuscript.

Thank you for the comments. In the revised manuscript, we have revised the conclusions and discussion to include the effect of material properties on nonlinearity as a focus of our future research. in response to the reviewer's comments related to seismic sources and model simplification, although these are not the focus of the current study, their significance for nonlinear studies cannot be denied, and therefore we have included them in the discussion as well. We appreciate the reviewer's feedback on enhancing the quality of our manuscript.

**Replies to Reviewer#2**

**1. Theoretical foundation**

Thank you for the comments. It made us realize that introducing too many equations and terms in the manuscript may confuse readers. Our original intention was to demonstrate the impact of nonlinear strain, but its presentation may have been too complicated. We have simplified the equations to avoid unnecessary repetition. At the same time, we will strengthen the explanation of nonlinear terms to clarify their physical meaning. We have thought about the reviewer's comments and it made us realize the mistake we made in the derivation part of the equations. There is no need to clarify and expand on the volumetric strain specifically, we have revised and rewritten this part in the revised manuscript and redone the simulation accordingly.

Based on the specific questions, we have the following responses:

**(1):** The rewriting of Eq. (2) as Eq. (5) in the manuscript is to show the approximate relationship between the nonlinear strain tensor $E_{ij}$ and the linear strain tensor $e_{ij}$ after neglecting the high-order nonlinear terms. We think this helps to understand the equivalence between $E_{ij}$ and $e_{ij}$ more clearly. In the revised manuscript, we still keep Eq. (5), retaining it to highlight the higher-order terms that are neglected in it in order to achieve the approximate with eij.

$$E_{ij} = \frac{1}{2}(\frac{\partial u_j}{\partial x_i} + \frac{\partial u_i}{\partial x_j} + \frac{\partial u_k}{\partial x_i} \cdot \frac{\partial u_k}{\partial x_j})  \tag{2}$$

$$E_{ij} = e_{ij} + \frac{1}{2}e_{ij}^2 + \frac{1}{2}(e_{ij}r_{ij} - r_{ij}e_{ij}) - \frac{1}{2}r_{ij}^2  \tag{5}$$

We have removed unnecessary expansions on volumetric strains and rederived the formulas and velocity-stress equations. Please see pages 6 – 10.

~~**(2):** Eqs (6), (7), and (8) are introduced to describe the volume changes that occur in a medium under nonlinear conditions when subjected to external forces. The volume strain includes the fundamental terms of the linear theory and the nonlinear terms. These neglected nonlinear terms also affect the propagation characteristics of elastic waves. Therefore, we introduce the volume strain here to preserve the nonlinear terms resulting from the volume strain, which are of the~~

same order as those in the nonlinear strain tensor, in order to better understand and simulate the propagation of elastic waves under nonlinear conditions. By retaining nonlinear terms of the same order in the volume strain as in the nonlinear strain tensor, not only the nonlinearities due to the strain tensor are taken into account, but also the nonlinearities due to the volume strain. We believe that the higher-order terms in the volume strain need to be extended and added to the nonlinear study. In future research, we will further discuss the difference in wave equations of motion considering only the nonlinear terms that the strain tensor brings to the propagation of elastic waves.

$$\theta = \frac{(1+\theta_{xx})(1+\theta_{yy})(1+\theta_{zz})dxdydz - 1dxdydz}{dxdydz} \tag{6}$$
$$= e_{xx} + e_{yy} + e_{zz} + e_{xx}e_{yy} + e_{xx}e_{zz} + e_{yy}e_{zz} + e_{xx}e_{yy}e_{zz}$$

$$\theta_e \approx e_{xx} + e_{yy} + e_{zz} = \frac{\partial u_x}{\partial x} + \frac{\partial u_y}{\partial y} + \frac{\partial u_z}{\partial z} \tag{7}$$

$$\theta_E \approx E_{xx} + E_{yy} + E_{zz} + E_{xx}E_{yy} + E_{xx}E_{zz} + E_{zz}E_{yy} \approx \frac{\partial u_x}{\partial x} + \frac{\partial u_y}{\partial y} + \frac{\partial u_z}{\partial z} + \frac{\partial u_x}{\partial x}\frac{\partial u_y}{\partial y} + \frac{\partial u_x}{\partial x}\frac{\partial u_z}{\partial z} + \frac{\partial u_y}{\partial y}\frac{\partial u_z}{\partial z} \tag{8}$$
$$+ \frac{1}{2}\left( \frac{\partial u_x}{\partial x}\cdot\frac{\partial u_x}{\partial x} + \frac{\partial u_y}{\partial x}\cdot\frac{\partial u_y}{\partial x} + \frac{\partial u_z}{\partial x}\cdot\frac{\partial u_z}{\partial x} + \frac{\partial u_x}{\partial y}\cdot\frac{\partial u_x}{\partial y} + \frac{\partial u_y}{\partial y}\cdot\frac{\partial u_y}{\partial y} + \frac{\partial u_z}{\partial y}\cdot\frac{\partial u_z}{\partial y} + \frac{\partial u_x}{\partial z}\cdot\frac{\partial u_x}{\partial z} + \frac{\partial u_y}{\partial z}\cdot\frac{\partial u_y}{\partial z} + \frac{\partial u_z}{\partial z}\cdot\frac{\partial u_z}{\partial z} \right)$$

**(3):** About Eq. (9) and Eq. (10): Eq. (9) is derived from the combination of linear strain tensor, linear volume strain, and related equations. This equation is the basic theoretical framework widely used in seismology in conventional linear theory. In contrast, Eq. (10) is obtained by combining the strain tensor, nonlinear volume strain, and related equations, and is designed to more comprehensively capture the nonlinear elastic effects that may occur during earthquakes, especially under strong or near-field earthquakes, which can significantly affect the characteristics and propagation patterns of seismic motion. By presenting the two equations in parallel, we want to visualize the essential differences between linear and nonlinear motion in mathematical expressions and their potential effects on seismic displacement field.

**2. Mathematical errors and confusion in equations**

Thank you for the comments. The mathematical errors and confusion in the equations

were indeed unintentional on our part. We have reviewed and corrected the use of symbols in the equations to ensure consistency. In particular, regarding the confusing use of i, j, k and x, y, z, we have standardized the symbol system used and provided clear definitions and explanations. Thank you for your detailed advice and patience. We will improve the equation derivation process and present it better.

**3. Benchmark required**

Thank you for the comments. We will validate our simulation results under linear approximation conditions by selecting a benchmark test code based on your suggestions. The correctness and reliability of our code will be verified by comparing the results of our code with the benchmark test code. We attempted to perform a benchmark using specfem3D, a tool we are not yet fully proficient with. The simulation and comparison we conducted (please see attached file) were quite basic and insufficient. We understand that further benchmark work is necessary and will pursue this to enhance the reliability of our research.

**4. English expression needs improvement**

Thank you for the comments. We acknowledge that there are shortcomings in our English expression that make it difficult to read. We will seek professional language services to improve the English expression and ensure the readability and academic standard of the manuscript. At the same time, we will improve our English language skills to ensure we can express our research results more clearly and accurately in our future research. We have revised the wave equations, conducted new simulations, and thoroughly reanalyzed the results. In the process, we have essentially rewritten the manuscript, ensuring we use professional and appropriate English academic language. Although this consumed a significant amount of time, it was necessary to improve the clarity and accuracy of our work. Unfortunately, due to time constraints, we could not seek professional language services before submitting our revisions. However, we will consider this option in the future to further enhance the linguistic quality of our manuscript.

In summary, we will address the theoretical foundation part by rewriting it according to the reviewers' comments, simplifying the equations, clarifying the derivation process, and removing unnecessary equations. For the numerical implementation, we will perform benchmark tests to ensure the consistency of the simulation results with known codes. We look forward to better presenting our research results in the revised version.

Thank you for your comments, especially for pointing out the critical error in our formula derivations. The comments were crucial in prompting us to re-examine and correct inaccuracies in our theoretical section. This was a significant issue for our research, and we sincerely appreciate the feedback in ensuring the accuracy of our work.

**List of major changes in the revised manuscript**

**1. Introduction**

1) Checked the citations and errors of references and revised the presentation of some content.

**2. Theory and method**

1) Revised and rewrote the derivation of the equations, ensured that consistent notation rules were used, and modified the notation in Figure 1 to match the text description.

2) Adopted reviewer comment #2's suggestion to modify and rewrite the derivation of the wave equations incorporating the nonlinear terms, with additional explanations of the physical meaning of the nonlinear terms. (Pages 6 - 7)

3) Improved the presentation of the staggered-mesh finite-difference method and rewrote the relevant parts of the velocity-stress equation. (Pages 8 - 9)

4) A description of the free-surface condition used in the simulation method was added. (Page 10)

**3. Simulations of basic seismic moment sources**

1) Rewrote the description of the seismic moment tensor source and the loading method of the simulated sources. (Pages 11 - 12)

2) Re-simulation based on the modified velocity-stress equation. The amplitude of the full-space model is adjusted from 6 to 7. The original model size of 60km×60km× 60km is now adjusted to 80km×80km×80km, and the seismic source is still located at the center.

3) Modified Figs. 3, 4, and 5 of the three basic source simulation results to show slices of the wavefield snapshots in all three planes for six components, where the error calculation of wavefield was removed because it is not reasonable to do the calculation directly in the wavefield snapshots; and rewrote the analysis of the simulation results for the three sources. (Pages 14 - 17)

4) Based on the simulation results after modifying the wave equations, the relative

change rate of wavefield energy with seismic magnitude (Figure 6) and time (added as Figure 7) was recalculated, and the analyses were rewritten. (Pages 18 - 19)

**4. Observations and simulations of two earthquakes**

1) Added the introduction related to the observation of the two Taiwan earthquakes and deleted the array subplot in Figure7 (now Figure 8) irrelevant to this study. (Pages 21)

2) Re-simulated E1 according to the adjusted wave equations, modified Figure 8 (now Figure 9), and rewrote the analyses. (Page 23 - 24)

3) Added a comparison of the time-frequency spectra of the linear and nonlinear simulations of E1 (added to Figure 10) and rewrote the analysis of Figure 10. (Page 24)

4) Re-simulated E2 according to the adjusted wave equations, modified Figure 9 (now Figure 11), and rewrote the analyses. (Page 26)

5) Added a comparison of the time-frequency spectra of the linear and nonlinear simulations of E1 (added to Figure 12) and rewrote the analysis of figure 12. (Page 27)

**5. Discussion and conclusion**

Based on reviewer comment #1's recommendation for revised manuscript analysis, we have rewritten the conclusion and the discussion sections, including summarizing the conclusions obtained from the simulation results of existing models and further discussing the necessary nonlinear simulation research work to be done in the future. (Pages 29 - 30)

---

## Referee Report (RR1)

**Review of the paper "Simulation characteristics of seismic translation and rotation under the nonlinearity in small deformation" by**

**Wei Li[1,2], YunWang[1,2], Chang Chen[1,2] and Lixia Sun[1,3]**

[1] *"MWMC" group, School of Geophysics and Information Technology, China, University of Geosciences, Beijing 100083, China*
[2] *State Key Laboratory of Geological Processes and Mineral Resources, China, University of Geosciences, Beijing 100083, China*
[3] *Sinopec Research Institute of Petroleum Engineering Co., Ltd., Beijing 102206, China*

The authors present an updated version of their work studying the non-linear effects in seismic wave propagation. This is a much clearer work and it is evident the large amount of efforts that the authors put on this updated version. Without stating whether they are correct or not, one can clearly read the motivation, the mathematics and the results. This means that now the paper has a clear structure, which did not have before.

In the next, I elaborate on each part of the structure. Before doing so, I have to acknowledge that the authors have done a monumental work, and despite this I will recommend that there is still work to do. This I understand, can be very frustrating and painful when one is writing this kind of work. However it is a process that, I believe, all of us that work in this line of research have to learn and to endure.

**1 The motivation of the work**

The motivation and importance of the work is clear now. The authors have added pertinent references and highlighted why this work is important to do and to publish. To the best of my knowledge, it has not been done before. So good on this side.

**2 The mathematics**

The very complicated mathematics has been removed, however there are still errors /modifications to be done in the equations.

- In eq. (9) the authors insist on using the symbol $\theta$ inside the equation. This on the one hand is very unusual and on the other hand is very confusing and mathematically inconsistent. The symbol $\theta$ is defined as the principal strain in the previous paragraphs. It is define in terms of the sub-indexes 1,2,3. Eq. (9) on the contrary is written in terms of $i, j, k$. So imagine now that I write Eq. (9) in terms of $i, j$ only. What happens to the definition of $\theta$ then? which ones are the two sub-indexes that are going to survive? can you see the inconsistency?

  I have to add that I do not see any need of writing eq. (9) using $\theta$, I find it confusing, inconsistent and additionally, nobody does it this way. So why the authors have to add this complexity into the very well known linear elastic equations? Please write it in the conventional way or in a consistent way, but keep in mind to keep it simple and understandable.

- What is $dt$ doing there in those equations of motion (eq. (11)) that have not yet been discretized?

- Can you please add with an underbrace which are the new terms included in the equations? In comparison to the linear elastic case. I mean, something like these are the old terms and these are the new terms. Use underbrace or use colors, blue for the linear elastic equation and red for the new terms. This will add clarity to the equations. You can also split the equation con continue in the next line. This will allow to write an equation with larger letters. Currently the equations of motion are very small in comparison to the main text.

[Figure]

Figure 1: Please replace by the 3D staggered grid instead of the 2D. The 3D figure is taken form the manual of the code FD3S.

**3 The results**

The results are interesting however they need a new presentation and the authors still need to, I am very sorry to say (because I do understand how painful this is), to validate the code.

**3.1 Validation of the code**

So let's start from the (most painful part) validation of the code first and then go to the easy part (the graphics). The authors include a plot of the not successful validation fo the code. Yes only people who work with specfem know how difficult this part is. The authors tried to directly benchmark their code with specfem and this is of course difficult to do (not impossible however). The point is simply, one cannot believe any results of a code that has not been benchmarked and the staggered grid discretization is not clearly explained.

In the next I offer three different paths that the authors can take, which in my opinion, are easier to do in comparison against learning how to properly use specfem3d.

- There are some folders inside specfem3d/EXAMPLES/ that give you the analytical solution. The authors can take this path and try to do a match of these analytical solutions found in the EXAMPLES folder.

- There is, in my opinion, an easier path. The authors can go to the SPICE Project webpage SPICE Project and download the package FD3S FD3S exactly here FD3S-here. The FD3S model seismic wave propagation in spherical coordinates using sponge absorbing boundary conditions and including viscoelasticity. It has been initially written by Prof. Heiner Igel. It is already benchmarked and most importantly it uses staggered grid in 3D. This comes because it is not clear how the authors do the staggered grid in 3D. There are numerous ways to do it: partly staggered, full staggered, so how the authors did it? The authors add a 2D discretization plot (Fig. 2) of a 3D wave equation. This is simply not acceptable because it says nothing about what the authors did. Pleas replace the 2D figure by the proper 3D staggered discretization (see Fig. 1).

  With the FD3S code the authors will see how the staggered grid is done in 3D, they can see how to implement future viscoelasticity and MPI parallel distribution with their own code (in case they want to further develop their C++ code). The authors can do a benchmark of their own code using this one. The simply need to install the code, run an example and make a conversion between Cartesian and Spherical coordinates. It is the authors decision however.

- The authors can also write Prof. Moczo (or his group) nuquake webpage and ask for help for the benchmark of the code.

**3.2   The simulations and graphical representations**

I have to say here that is very clear the nice effort done in trying to explain everything. It is a nice work. However, please understand that one cannot believe any result if the code has not been properly benchmarked. But I do acknowledge the effort done by the authors. Next are my suggestions.

First of all the authors need to clarify the number of points per wavelength that they are using (and why this selection) and the Courant number. Next, the authors present several plots that are nice but not very informative, for example, Figs. 3 to 5 are OK but they are qualitative plots, they serve only as a qualitative analysis of a certain time step during the simulations. So, I do not really know how much effort should be put on a certain snapshot and in this kind of analyses.

Next, for the time series comparisons, those plots are very blurry. This is just a technical detail yes, but it is important. Plots need to be clear in all possible ways. Numbers should be readable, the plot should not be blurry and they have to be scientifically informative.

The data comparison is a very ambitious Section. The authors have a code that it has not properly been benchmarked, does not include attenuation, does not include the Earth curvature, the earth model included is 1D layered and they want to compare to data. Ok that is ambitious. It can be done but if the authors properly discuss the limitations of the physics of the comparison, which they clearly try to do. That is very nice but the results are shown in not the most informative way.

Every observational seismologist would like to see the three components separated and rotated to Transverse, Radial and Vertical directions and not XYZ. This is crucial, because in the Transverse component one should not see a P wave for example. In the Radial and Vertical component one should see the interaction between P and S waves and it is only then that one can really talk about the physics of rotations and compare to observations. Otherwise, those separated XYZ plots are very little informative and basically all theoretical work is not properly used.

I will stop here. I think it is more than enough corrections.

**4   Summary**

I do believe that the authors have made a significant amount of work but the paper is not yet ready for publication. The reason is because the papers is just too ambitious. Basically the authors are writing a 3D code from scratch, doing a theoretical analysis and a data comparison. This are three papers:

1. The publication of the code properly benchmarked and explained and with a manual (made publicly available).

2. The publication of the theoretical analysis: the authors can use the code for doing synthetic calculations in 1D models and analyzing results, then changing to a 3D model and analyzing results and then using hypothetical 3D models to see the effects on the rotations and analyzing results.

3. The publication of the data and simulation comparisons: Now the authors will have an idea of the results to expect from the second paper, then they can go directly to talk about the physics of the observations and how the non-linearities are properly taken into account or not.

On the contrary, the authors put these three papers into a single paper making a ridiculously amount of work for the author (or person) who is in charge of the project and for the reviewers. Of course, basics things like testing of the code, availability, benchmark and all details previously mentioned are not going to be covered because it is simply too much work for a single paper. In addition, for the first two papers one has to think as a theoretician but for the last paper as an observational scientist. Those are two completely different ways of using our brains and it is an additional challenge.

In any case, I can continue writing but I will stop here. The authors can do what they consider best and more appropriate for them. I simply hope these lines help the authors to improve and make easier their (current and future) work.

A reviewer

---

## Referee Report (RR2)

Review of the paper:

**Nonlinear Wavefield Characteristics of Seismic Translation and Rotation in Small-Strain Deformation: Insights from Moment Tensor Simulations**

Wei Li [1,2], Yun Wang [1,2,], Chang Chen[1,2] , Lixia Sun [1,3]

1 "MWMC" group, School of Geophysics and Information Technology, China University of Geosciences, Beijing 100083, China
2 State Key Laboratory of Geological Processes and Mineral Resources, China University of Geosciences, Beijing 100083, China
3 Sinopec Research Institute of Petroleum Engineering Co., Ltd., Beijing 102206, China

The authors present an updated version of their previous submitted work. Despite the topic is of interest to the community I have rejected the paper in several previous occasions and despite the authors have made some modifications to the text, I am sorry but I still recommend rejection of this work. I will elaborate why I still think this way.

As I previously stated, the authors present a numerical study based on a code that they have developed. This is something important and difficult to do. However, they don't do any benchmark of their code, which makes the reader (very) doubtful of their numerical results.

On the one hand, one can see the effort done by the authors on improving the clarity of the presentation of the mathematics and the structure of the paper. This is nicely done now. The paper has a clear structure and the mathematics is clearly presented. On the other hand, the lack of validation of the code renders their numerical experiments doubtful.

I can bypass, for a moment, the lack of benchmarking of their code and trust that the numerical code is well written and read and try to understand their simulations. Then it comes the next problem: The authors present an analysis of seismograms (time histories) and they insist in plotting all components over the same line and present, in my opinion, a mathematical/statistical study of the waveforms. This, in my onion, is not a proper way to understand seismic waveforms because: on the one hand we look to understand the physics of wave propagation when we analyze seismograms and, on the other hand, this kind of presentation obscures the information that seismograms contain.

In simpler words: when we look at differences in seismograms we look to analyze differences in different waveforms: P waves, S waves, Love and or Rayleigh waves, etc and not to say this part of the seismogram is different and the other one not. This will tell you the properties of the material where the non-linearity is observed and why is it this way.

An easy way to invalidate the results presented in this paper is by looking at Figs. 11 and 12. If one makes a zoom we can see the similarity between trans.R and trans.T at the beginning of the seismogram and this simply cannot be happening in an isotropic medium. The first arrival that we are able to see in trans.R and trans.Z is the P wave, however, trans.T should not show any arrival whatsoever. The first arrival of the trans.T component is the S wave an before that one should see simply nothing, absolutely zeros. This is seismology 101.

One then is left with the doubt: the authors did not correctly benchmark their code? and/or their simply fail in rotating the seismograms? I cannot answer this question from the information

presented in the manuscript. Therefore, I am sadly to say that I cannot believe their results in any case.

I do acknowledge that the authors present a challenging study but if there is the inability of benchmarking their code in 3D and there are serious doubts about their results, I canot believe what is written here. I thus suggest to simplify the study. The authors could simply run a 2D SH wave simulation to which the analytical solution is very easy to do for the code (see Heiner Igel's book) and the authors can then focus on understanding the difference in their seismograms in different 2D SH heterogenous scenarios. In another paper the authors can focus on 2D PSV and repeat the experiment. This will allow to gain insight and intuition on how to analyze seismograms and numerical model the results.

To make the review short, I am sorry to say that I still find hard to believe the results presented by the authors. They have improved a lot; however, I am still left with serious doubts about the validity of the results presented here because the lack of validation of the code and due to the incongruence in the physics that the seismograms show.

With best regards,

A reviewer

---

## Referee Report (RR3)

Review of the paper:

**Nonlinear Wavefield Characteristics of Seismic Translation and Rotation under the nonlinearity in Small-Strain Deformation: Insights from Moment Tensor Simulations**

Wei Li [1,2], Yun Wang [1,2,], Chang Chen[1,2] , Lixia Sun [1,3]

1 "MWMC" group, School of Geophysics and Information Technology, China University of Geosciences, Beijing 100083, China
2 State Key Laboratory of Geological Processes and Mineral Resources, China University of Geosciences, Beijing 100083, China
3 Sinopec Research Institute of Petroleum Engineering Co., Ltd., Beijing 102206, China

The authors present an updated version of their previous submitted work. Despite the topic is of interest to the community I have rejected the paper in several previous occasions.

The authors have carefully done several modifications to their work and I acknowledge that. compared to the initial version of this work, the authors have considerably increased the quality of research and presentation of their work.

The presented work has a very clean and clear methodological structure now. It is easily readable and the message is clear. I however have some major comments. This time is not major in the sense of methodological structure (like previous revisions) and/or agree or disagree with what is written there, but some important points that need to be addressed.

My major comments are only few:

(1) If the authors mention the simulation of a realistic event. Then I have to see data comparison. After the theoretical work done, which is credible, it makes no sense to talk about an important event and not compare to data. This will allow the reader to validate the methodology and importance of the work from the data point of view. This is important if the author insist on modeling a realistic event. One can simply download the data from IRIS for this event and after a little waveform processing, do the data comparison. If the authors are not familiar with this kind of data processing, please visit the Obspy tutorials.
The main idea of data comparison is not to state whether your numerical modeling reproduces the data. The authors are not doing a tomography. The comparison instead will help to have an idea how accurate the 1D non-linear structural model reproduces the data and to check whether a better data comparison can be achieved by the non-linear model. It does not have to state that the non-linear model reproduces better the data compared to the linear one (of course this will be the ideal scenario). The main idea is to analyze whether seismograms can be reproduced by a non-linear model (or not). It is just reporting the whether this can be achieved (or not) with the current assumptions and to discuss the reasons.

(2) The authors need to be aware that the term non-linearity is used in the literature to describe many different physical effects. For example, the term non-linearity can be used to describe visco-elastic plastic effects (Bonilla et al. 2005, Bonilla et al. 2011, Frankel et al. 2011, Régnier et al. 2018). However, in this study, non-linearities are geometrical, to describe pre-stressed medium (see Tromp & Trampert 2018 and

references therein). These are two completely different things. So, I would like to read a discussion in the paper clarifying these differences to the reader.

In addition, since the geometrical non-linearity is used to describe pre-stressed media (see Tromp & Trampert 2018), how is this related to the medium when the realistic earthquake is modeled? Is it realistic to do this? Why so? This is very important to discuss.

(3) The benchmark tests should be more explicitly explained inside the paper. Even if it is written as a supplementary information or as an appendix. There must be a discussion of the benchmark done to the code.

(4) Finally, the presentation of the time histories (seismograms) is not the most appropriate one (in my opinion). I still lack of a clear comparison of the transverse component. Simply because the transverse component should not have any energy before the S wave and I cannot see that in the plots. It seems tricky to me how the authors are rotating the seismograms. While this is a minor comment, it is important to clearly show and present in the text.

I would like to respectfully acknowledge the large amount of work that the authors have done at this point. I do believe that after the previous mentioned modification are done, this can easily be a very cited and valuable work.

With best regards,

A reviewer

**REFERENCES**

Bonilla, L. F., Archuleta, R. J., & Lavallée, D. (2005). Hysteretic and dilatant behavior of cohesionless soils and their effects on nonlinear site response: Field data observations and modeling. Bulletin of the Seismological Society of America, 95(6), 2373-2395.

Bonilla, L. F., Tsuda, K., Pulido, N., Régnier, J., & Laurendeau, A. (2011). Nonlinear site response evidence of K-NET and KiK-net records from the 2011 off the Pacific coast of Tohoku Earthquake. Earth, planets and space, 63(7), 785-789.

Frankel, A. D., Carver, D. L., & Williams, R. A. (2002). Nonlinear and linear site response and basin effects in Seattle for the M 6.8 Nisqually, Washington, earthquake. Bulletin of the Seismological Society of America, 92(6), 2090-2109.

Régnier, J., Bonilla, L. F., Bard, P. Y., Bertrand, E., Hollender, F., Kawase, H., ... & Verrucci, L. (2018). PRENOLIN: International benchmark on 1D nonlinear site-response analysis—Validation phase exercise. Bulletin of the Seismological Society of America, 108(2), 876-900.

Tromp, J., & Trampert, J. (2018). Effects of induced stress on seismic forward modelling and inversion. Geophysical Journal International, 213(2), 851-867.

---

## Referee Report (RR4)

Review of the paper:

**Nonlinear Wavefield Characteristics of Seismic Translation and Rotation in Small-Strain Deformation from Moment Tensor Simulations**

Wei Li [1,2], Yun Wang [1,2,], Chang Chen[1,2] , Lixia Sun [1,3]

1 "MWMC" group, School of Geophysics and Information Technology, China University of Geosciences, Beijing 100083, China
2 State Key Laboratory of Geological Processes and Mineral Resources, China University of Geosciences, Beijing 100083, China
3 Sinopec Research Institute of Petroleum Engineering Co., Ltd., Beijing 102206, China

The authors present an updated version of their previous (long time) submitted work. Despite the topic is of interest to the community, I have rejected the paper in several previous occasions. This time, however, the paper has been substantially improved and clarified. The message of the paper is quite clear now and the contribution is new. The equations have been reduced to the minimum while keeping clarity and correctness on the presentation. Thank you for that.

This time, I only have minor suggestions that, in my opinion, will improve the clarity of the contribution:

Line 30: excitation efficiency. What is that?

Lines 195 and 196: it is not fully correct that when changing the waveform, one does not change the travel time. In many applications one measures travel times using envelopes, i.e., the maximum of the envelope, because measuring the beginning of the waveform is practically impossible to determine and cross-correlation measurements only work on similar waveforms. This means that the maximum of the envelope is a good indicator of waveform travel times. Please rephrase the sentence of lines 195-196 considering these notes.

Line 279: what is rectilinearity? Please define it.

Line 285: Same here, what is the Spearman's rank correlation coefficient. Please explain with clarity what it is and the reason of this choice.

Line 326: What do you mean by mixed sources? What is that?

Line 331: What is this time-frequency energy? Hilbert transform frequency, i.e., instantaneous frequency or Fourier frequency? Please clarify. In addition, please define inside the text that time-frequency-energy difference plots are the non-linear minus the linear. It is only defined in the figure caption.

Line 368: Fundamental seismic sources?

Please remove the part of specfem2d simulations. One cannot compare a 3D code with a 2D. The physics of wave propagation in 2D is different from 3D. One can never reach any fit comparison with different physics of wave propagation. Please remove this part completely and do not mention in the manuscript any 2D comparison. It makes no sense.

The benchmark with sismowine is fine, that gives the validity to the code that needed it. I, however, can see that the code of the authors shows some high frequency components in their waveform. This is usually normal and, in my experience, can be simply removed by filtering, so please do that. It should be ok then.

The Discussion is very informative and clearly written and Conclusions very clear too. The contribution of the work is clear and scientific. To the best of my knowledge, there is no other paper like this one published.

I thank the authors for the long and painful work done. I am sorry that it has been this painful. I do believe that now is a good and informative theoretical paper that gives a new message and motivates the community of rotational seismology to push forward.

With best regards,

A reviewer

---

## Author Response (AR2)

**Nonlinear Wavefield Characteristics of Seismic Translation and Rotation in Small-Strain Deformation: Insights from Moment Tensor Simulations**

Wei Li [1,2,], Yun Wang [1,2,*], Chang Chen[1,2], Lixia Sun[1,3]

[1] *"MWMC" group, School of Geophysics and Information Technology, China University of Geosciences, Beijing 100083, China*

[2] *State Key Laboratory of Geological Processes and Mineral Resources, China University of Geosciences, Beijing 100083, China*

[3] *Sinopec Research Institute of Petroleum Engineering Co., Ltd., Beijing 102206, China*

[*] Corresponding author: wangyun@mail.gyig.ac.cn.

**Replies to Reviewer Comments**

We extend our heartfelt gratitude for the reviewer's review and invaluable suggestions, which have enhanced the quality of our manuscript. We have thoroughly addressed each of the reviewer's comments and revised the manuscript accordingly. Below is a detailed account of our revisions and responses to the reviewer's feedback.

**1. Revisions to Equations and Mathematical Notation**

Response: Thank you for the comments. We have systematically revised the equations throughout the manuscript. Specifically, we removed the symbol $\theta$ from Eq. (9) and replaced it with standard tensor indices i,j,k, ensuring consistency with definitions in the text. For the velocity-stress equations (now Eq. (11)), we clarified the physical meaning of the time step *dt* and explicitly described its role in the discretization process (please see Lines 147-149).

Additionally, nonlinear terms in the equations are now highlighted using underlines and annotations to improve readability in the revised manuscript.

**2. Code Validation and Benchmarking**

Response: Thank you for the comments. We have replaced the original 2D staggered-grid schematic with a 3D diagram to clearly illustrate the spatial distribution of variables within the computational grid (please see Fig. 2). Following the recommendation, we conducted benchmark tests using the SPICE code framework. By comparing numerical results with reference solutions in homogeneous media models, the quantified waveform misfit errors, with detailed validation procedures and outcomes documented in the attached Accuracy Report, please find them attached.

**3. Optimization of Results Presentation**

Response: Thank you for the comments. In the "Forward modeling parameters" section, we added explicit criteria for selecting points per wavelength and the Courant number, ensuring numerical stability (Please see Lines 196-203). The single waveforms have been converted from the XYZ coordinate system to the RTZ system to align with observational seismology practices. We have streamlined lengthy qualitative descriptions and introduced quantitative comparisons, such as wavefield differences; please see the revised manuscript.

**4.Limitations of Data Comparisons**

Response: Thank you for the comments. In the revised "Discussion" section, we explicitly outline the current model's limitations, including the omission of attenuation, anisotropy, and small-scale heterogeneities. We further delineate future directions to address these constraints, such as incorporating viscoelasticity and complex media models; please see the revised manuscript.

**5. Restructuring and Phased Research Recommendations**

Response: Thank you for the comments. We fully understand the rationality of publishing the research in stages and have revised the manuscript. The revised manuscript focuses on the theoretical construction of nonlinear seismic wave propagation and the simulation and analysis of earthquake source mechanisms.

The comparison of observational data in the original manuscript has been deleted to ensure the focus of the topic, and we have included the nonlinear study of observational data as a possible independent and more in-depth and targeted research content in the future.

Regarding the computational code utilized in this study, we wish to clarify that the current manuscript does not prioritize code publication, as the codebase has not yet reached the standards required for public release. We are systematically refining the code architecture—including enhancing scalability features such as parallelization and viscoelastic attenuation modules—to ensure robustness and reproducibility. The present work focuses on theoretical advancements in nonlinear seismic wave propagation, while code dissemination and data-driven validation will be addressed in subsequent phases of this research program. We sincerely appreciate the reviewer's suggestions on improving the code ecosystem and will integrate these insights into our ongoing development efforts.

We deeply appreciate the reviewer's feedback on our manuscript and the guidance on code validation for elevating the rigor and clarity of this work. All revisions are highlighted in the revised manuscript, and please find the attached sumplement materials about the benchmark.

**List of major changes in the revised manuscript**

**1.** Rewritten the abstract and title.

**2. Theory and method**

Revised the equations, ensured that consistent notation rules and symbols were used in all equations, and simplified the presentation of lengthy tensor formulas. (Pages 6 - 9)

**3. Simulations of basic seismic moment sources**

1) Added quantitative analyses (Figs. 4, 6, 8) of the three source simulations and rewritten the analyses to remove unnecessarily lengthy qualitative analyses and to focus on the linkages to the force source mechanisms and quantitative analyses. (Pages 12 - 18)

2) Modified Fig.6 (now Fig. 9), with the addition of a sensitivity comparison of rotational and translational components, rewritten the analysis of the simulation results (with added Table 1), and deleted the comparison results over time. (Pages 19 - 20)

**4. Observations and simulations of two earthquakes**

Deleted the presentation of measured data irrelevant to this study objective, focusing on modeling and analyzing source-related nonlinear effects recorded by ground motions of the two earthquakes, and rewritten the analysis content. **(Pages 21 - 25)**

**5. Discussion and conclusion**

Based on the modified above analysis content, we have rewritten the conclusion and the discussion sections. (Pages 26 - 28)

---

## Author Response (AR4)

**Responses to Reviewers**

**Responses to Reviewer #1' Comments**

1. As I previously stated, the authors present a numerical study based on a code that they have developed. This is something important and difficult to do. However, they don't do any benchmark of their code, which makes the reader (very) doubtful of their numerical results.

   ...

   On the other hand, the lack of validation of the code renders their numerical experiments doubtful.

   Thanks for the reviewer's suggestions.

   We have added a Supplementary Material with benchmarking results, including comparisons with SPICE/SISMOWINE and SPECFEM2D:

   a. Comparison with the SPICE/SISMOWINE international standard program: We selected the standard test cases from this project and compared our simulation results with the provided reference solutions through waveform comparison and quantitative misfit analysis. The results show that our code's travel times, amplitude envelopes, and overall waveforms are in good agreement with the reference solutions.

   b. Validation with the SPECFEM2D spectral-element program: We also performed a cross-validation using the widely recognized open-source spectral-element package SPECFEM2D. We compared the waveforms from both codes under full-space and half-space models, respectively. The results again showed good consistency.

2. The authors present an analysis of seismograms (time histories) and they insist in plotting all components over the same line and present, in my opinion, a mathematical/statistical study of the waveforms. This, in my opinion, is not a proper way to understand seismic waveforms because: on the one hand we look to understand the physics of wave propagation when we analyze seismograms and, on the other hand, this kind of presentation obscures the information that

seismograms contain. In simpler words: when we look at differences in seismograms we look to analyze differences in different waveforms: P waves, S waves, Love and or Rayleigh waves, etc...

Thanks for the reviewer's suggestions.

We have recognized that the analysis of waveforms in the previous version was overly reliant on general statistical metrics and lacked an in-depth analysis of wave phases and physical mechanisms. Based on the reviewer's suggestion, we have revised the result analysis framework to focus on wave phases and physical mechanisms (please see Section 3 especially).

3.  An easy way to invalidate the results presented in this paper is by looking at Figs. 11 and 12. If one makes a zoom we can see the similarity between trans.R and trans.T at the beginning of the seismogram and this simply cannot be happening in an isotropic medium. The first arrival that we are able to see in trans.R and trans.Z is the P wave, however, trans.T should not show any arrival whatsoever. The first arrival of the trans.T component is the S wave an before that one should see simply nothing, absolutely zeros. This is seismology 101.

Thanks for the reviewer's suggestions.

We reviewed our data processing procedure and discovered an error in calculating the azimuth angle when rotating the simulation results from the Cartesian coordinate system to the RTZ coordinate system. We have corrected this algorithm. In the revised manuscript, the seismograms shown in the RTZ coordinate system in the corresponding Figs. 8 and 9 have been processed with the correct coordinate rotation.

4.  I thus suggest to simplify the study. The authors could simply run a 2D SH wave simulation to which the analytical solution is very easy to do for the code... and the authors can then focus on understanding the difference in their seismograms in different 2D SH heterogenous scenarios.

Thanks for the reviewer's suggestions.

Regarding the specific path the reviewer recommended, we would like to explain from two aspects:

a. Regarding code validation: During the revision period, we attempted to follow the 2D SH validation path. However, we encountered some technical difficulties in obtaining and implementing the relevant analytical solutions and could not complete it successfully. Therefore, we compared our code with the 3D standard model of the SPICE/SISMOWINE project and performed a cross-validation of the 2D P-SV wavefield with the widely used SPECFEM2D program. The relevant comparison results are presented in the supplementary material we submmited.

b. Regarding simplifying the research: After consideration, we decided to retain the current 3D model research framework. This is mainly because we want to investigate the spatial distribution characteristics of nonlinear effects excited by three-dimensional source mechanisms. The points we are interested in, such as how the efficiency of nonlinear mode conversion is controlled by the source radiation pattern, and the characteristics of the rotational motion itself, including all its components, exhibit anisotropic behavior in three-dimensional space. If the model were simplified to 2D, we would only be able to observe phenomena on a specific profile, which would inevitably lead to a incomplete understanding of the spatial characteristics. Therefore, to capture and analyze these phenomena with 3D properties, we retain the 3D simulation framework.

**Responses to Reviewers #2' Comments**

1.  If the authors mention the simulation of a realistic event. Then I have to see data comparison. After the theoretical work done, which is credible, it makes no sense to talk about an important event and not compare to data. This will allow the reader to validate the methodology and importance of the work from the data point of view... The main idea is to analyze whether seismograms can be reproduced by a nonlinear model (or not).

    Thanks for the reviewer's suggestions.

    We agree with the reviewer's viewpoint that comparing simulation results with real observational data is the powerful means of validating a theory and method. However, at the current stage of this research, we plan to perform data comparison in future work to validate with real records, as suggested. Because the primary objective of this study is to isolate and understand the effects produced by the specific mechanism of geometric nonlinearity within a controlled numerical experiment environment. To achieve this goal, even when simulating the referenced earthquake events, we only used a simplified, horizontally layered medium model to exclude interferences from factors such as complex 3D structural heterogeneity, small-scale scattering, and site effects.

    We position this study as a fundamental theoretical and numerical investigation, and the suggestion represents a crucial next step for this research. In our future work, we will attempt to apply our nonlinear model to real data and analyze whether the nonlinear model can better explain the observed waveforms. We have also pointed this out in the Discussion part, identifying it as a future research direction.

2.  The authors need to be aware that the term nonlinearity is used in the literature to describe many different physical effects. For example, the term nonlinearity can be used to describe visco-elastic plastic effects... However, in this study, nonlinearities are geometrical, to describe pre-stressed medium... I would like to read a discussion in the paper clarifying these differences to the reader.

    Thanks for the reviewer's suggestions.

We have added a clear distinction and explanation of different nonlinear mechanisms in the Introduction. In the revised manuscript, we explicitly state that the scope of this study is limited to geometric nonlinearity and is conducted within a linear elastic constitutive framework, avoiding confusion with studies on material nonlinearity.

3. The benchmark tests should be more explicitly explained inside the paper. Even if it is written as a supplementary information or as an appendix. There must be a discussion of the benchmark done to the code.

Thanks for the reviewer's suggestions.

We completely agree with the reviewer's suggestion. We have prepared a Supplementary Material dedicated to presenting the benchmarking results of our code. In the supplementary material, we have compared our code with the SPICE/SISMOWINE international standard program and the SPECFEM2D spectral-element program, covering both full-space and half-space models in waveform comparison plots and quantitative error analyses. Please see the supplementary.

In Section 2.2 (Staggered-grid ginite-difference method), we have explicitly informed the readers that we have conducted this validation work and directed them to the supplementary material for detailed information.

4. Finally, the presentation of the time histories (seismograms) is not the most appropriate one (in my opinion). I still lack of a clear comparison of the transverse component. Simply because the transverse component should not have any energy before the S wave and I cannot see that in the plots. It seems tricky to me how the authors are rotating the seismograms.

Thanks for the reviewer's suggestions.

It's correct that in an isotropic or horizontally layered medium, the P-wave first arrival should not appear on the transverse component. After careful inspection, we found that an error occurred when rotating the simulation results from the Cartesian coordinate system to the RTZ coordinate system in the previous version. We have since corrected the coordinate rotating algorithm. In the revised

manuscript, all figures showing RTZ components (Figs. 8 and 9) have been correctly processed. The transverse component is at the numerical noise level before the S-wave arrival, which is consistent with basic wave physics principles.

**List of relevant changes made in the manuscript**

We have made substantial revisions to the manuscript based on the reviewers' comments. The main changes are listed below:

**I. Major methodological and structural revisions**

1. Addition of code validation

In response to the critical concern about the lack of code validation (Reviewer #1, Comment 1.1; Reviewer #2, Comment 2.3), we have prepared a Supplementary Material dedicated to the benchmarking of our numerical code. This includes detailed waveform and misfit comparisons with two international standard benchmarks: the SPICE/SISMOWINE project and the SPECFEM2D package. A statement directing the reader to this material has been added to Section 2.2.

2. Correction of a fundamental physical error in seismograms

Reviewer #1 (Comment 1.3) and Reviewer #2 (Comment 2.4) correctly pointed out a fatal error in our previous results where P-wave energy appeared on the transverse (T) component. We have identified and corrected the error in our coordinate rotation algorithm (from Cartesian to RTZ). All figures showing RTZ seismograms (Fig. 8 and Fig. 9) have been re-processed and replaced. The corrected figures show no energy on the T component before the S-wave arrival, which is consistent with basic wave physics.

3. Complete revision of the analysis framework

As suggested by Reviewer #1 (Comment 1.2), we have shifted the focus of our analysis from a general statistical study to an in-depth investigation of physical mechanisms. The entire Chapter 3 has been restructured to analyze nonlinear effects in terms of their spatial radiation patterns, spectral characteristics, and their impact on micro-waveform properties (e.g., rectilinearity and rotation-translation correlation), with an emphasis on the physical interpretation of the results.

**II. Revisions in specific sections**

1. Introduction: Following the advice of Reviewer #2 (Comment 2.2), we have added
    a clear distinction between constitutive nonlinearity and geometric nonlinearity

in the ssecond and third paragraphs of the Introduction to clarify the scope of our study.

2. Results and Discussion (Chapters 3, 4, 5): The entire results and discussion sections have been rewritten to reflect our deeper physical analysis.

3. Conclusions: The conclusions have been completely rewritten to be more precise and strictly aligned with the results. We now clearly distinguish between the universal radiation pattern and the source-dependent excitation efficiency of the nonlinear effects, and have specified that the high sensitivity of rotation is primarily for S-wave-driven nonlinearity.

---

## Author Response (AR5)

**Responses to Reviewer's Comments**

We would like to express our sincere gratitude for the reviewer's review and comments on our manuscript. We have revised the manuscript accordingly. Below are our point-by-point responses.

1. Line 30: excitation efficiency. What is that?

   Thanks for the reviewer's suggestions.

   We have replaced "excitation efficiency" with a clearer phrase: "the overall efficiency of exciting these effects". Please see Line 22.

2. Lines 195 and 196: it is not fully correct that when changing the waveform, one does not change the travel time... This means that the maximum of the envelope is a good indicator of waveform travel times. Please rephrase the sentence of lines 195-196 considering these notes.

   Thanks for the reviewer's suggestions.

   We have rewritten the corresponding sentences in the manuscript. We now explicitly state that the primary effect of nonlinearity is to modulate the wave's envelope shape, rather than perturbing its fundamental propagation phase. Please see Lines 198-199.

3. Line 279: what is rectilinearity? Please define it.

   Thanks for the reviewer's suggestions.

   We have added a clear definition for rectilinearity in the revised manuscript (Lines 285-287).

4. Line 285: Same here, what is the Spearman's rank correlation coefficient. Please explain with clarity what it is and the reason of this choice.

   Thanks for the reviewer's suggestions.

   We have expanded the explanation for Spearman's rank correlation coefficient in Lines 295-298.

5. Line 326: What do you mean by mixed sources? What is that?

   Thanks for the reviewer's suggestions.

   We have revised this term to describe the source mechanisms as composite, as they contain both double-couple and significant non-double-couple components. Please see Line 339.

6. Line 331: What is this time-frequency energy? Hilbert transform frequency, i.e., instantaneous frequency or Fourier frequency? Please clarify. In addition, please define inside

the text that time-frequency-energy difference plots are the non-linear minus the linear. It is only defined in the figure caption.

Thanks for the reviewer's suggestions.

We have changed "time-frequency energy" to the more accurate term "time-frequency spectrum" and clarified that these spectra are calculated by applying the Generalized S-Transform (GST) to the waveforms. Additionally, we have now defined in the revised manuscript that the time-frequency spectral differences are the result of the nonlinear simulations minus the linear simulations. Please see Lines 344-347.

7. Line 368: Fundamental seismic sources?

Thanks for the reviewer's suggestions. We have replaced the original phrase with a more specific description of the three basic moment-tensor source types.

8. Please remove the part of specfem2d simulations. One cannot compare a 3D code with a 2D. The physics of wave propagation in 2D is different from 3D... Please remove this part completely and do not mention in the manuscript any 2D comparison. It makes no sense.

Thanks for the reviewer's suggestions. We have removed the content and mentions related to the comparison with SPECFEM2D from both the supplementary material and the manuscript.

9. I, however, can see that the code of the authors shows some high frequency components in their waveform. This is usually normal and, in my experience, can be simply removed by filtering, so please do that. It should be ok then.

Thanks for the reviewer's suggestions. We have applied an appropriate low-pass filter to the waveforms shown in Fig. 8 and Fig. 9.

**List of relevant changes made in the revised manuscript**

In addition to the specific responses to the reviewer's comments above, we have summarized the revisions made to the manuscript as follows:

1. We have removed the comparison with SPECFEM2D simulations from both the manuscript and the supplementary material, as suggested by the reviewer, to avoid invalid comparisons.

2. We applied a low-pass filter to the synthetic waveforms of the referenced earthquake events (Figures 8 and 9) to remove high-frequency components.

3. We refined several terms for accuracy, including changing "excitation efficiency" to "efficiency of exciting these effects" and renaming "mixed sources" to "composite sources".

4. We added explicit definitions for "rectilinearity" (Lines 285-287) and "Spearman's rank correlation coefficient" (Lines 295-298) to better explain the waveform attributes used in the analysis.

5. We clarified the method used for time-frequency analysis (Generalized S-Transform) and explicitly defined the difference plots as "nonlinear minus linear" simulations.

6. We have updated the figure and table citations in the Supplementary Material to ensure consistency with the revised manuscript.